



# Past and future changes in avalanche problems in northern Norway estimated with machine-learning models

Kai-Uwe Eiselt[1] and Rune Grand Graversen[1,2]

[1]Department of Physics and Technology, University of Tromsø, Norway
[2]Norwegian Meteorological Institute, Tromsø Office, Norway

**Correspondence:** Kai-Uwe Eiselt (kai-uwe.eiselt@uit.no)

**Abstract.** Snow-avalanche hazard in mountainous areas may change in frequency and severity due to climatic change, especially in Arctic regions such as northern Norway experiencing Arctic temperature amplification. Building on earlier work, we train machine-learning models on dynamically downscaled reanalysis and model future projection data including snow-cover simulations to predict a binary avalanche danger metric (avalanche day/non-avalanche day) for the Troms county in northern Norway. Due to incomplete avalanche observations, we construct the metric from the avalanche danger warnings published in the Norwegian avalanche bulletin. The frequency of avalanche days is hindcasted for the period 1970 to 2024 (reanalysis) and projected into the future for the 21st century (climate model simulations). The results confirm earlier studies showing that while multi-decadal linear trends are marginal, the interannual variability of the avalanche-day frequency is linked to the Arctic Oscillation. The projected future changes indicate a general decrease of avalanche danger, especially for dry-snow avalanches. In contrast, wet-snow avalanche danger exhibits changes dependent on elevation, increasing at all elevations until mid-century, but thereafter continuing the increase only at higher elevation, while at lower elevation a decrease sets in. Our results are in line with an emerging consensus of a general decline of avalanche danger in the 21st century, however showing a shift in avalanche characteristics towards fewer dry and more wet-snow avalanches.

## 1 Introduction

A change in climatic conditions as observed during the 20th and as projected for the 21st century is likely to impact the occurrence and character of natural hazards. This is especially true for the types of natural hazards that are strongly influenced by meteorological parameters, as for instance snow avalanches. In the mountainous regions of Arctic Norway, snow avalanches are one of the most important natural hazards, threatening life, property, and infrastructure. According to data acquired from Norway's national mass-movement database (Nasjonale Skredhendelsesdatabasen, NSDB), from 1748 to the winter season 2024/25 snow avalanches have caused 354 fatalities in the northern-Norwegian county of Troms as shown in Fig. 1. Given the likely accelerated climate change in the Arctic due to Arctic amplification (i.e., the Arctic warming at a higher pace than the global average; e.g., Manabe and Stouffer, 1980; Graversen et al., 2008; Serreze et al., 2009; IPCC, 2021), enhanced impacts on snow avalanches in this region relative to, for instance, the Alps are to be expected.





Generally, the effects of climate change on snow avalanches appear to result in a decrease of the avalanche danger, i.e., decreasing number, size, seasonality, and active paths of avalanches, as summarised in the recent comprehensive review by Eckert et al. (2024). However, they also noted that increased snowfall at higher elevation may lead to increased snow-avalanche activity there, and that a shift is likely from dry to wet avalanches due to increased warming. Despite these rather clear effects, Eckert et al. (2024) observed that historical trends of avalanche frequency remain "elusive" (see also Førland et al., 2007; Sinickas et al., 2016; Gądek et al., 2017; Hao et al., 2023; Eiselt and Graversen, 2025) due to avalanche cycles and strong decadal variability. Negative historical trends have been found for the Swiss Alps (Teich et al., 2012), the French Alps (Eckert et al., 2013), and the Rocky Mountains (Peitzsch et al., 2021), while for the Himalayas (Ballesteros-Cánovas et al., 2018), and the Tianshan Mountains in China (Hao et al., 2023) positive trends were reported (the latter being not statistically significant). As noted by Eckert et al. (2024), the different trends are reconciled by elevation dependence and the simultaneous rise in temperature and precipitation. That is, warming and more liquid precipitation at low elevations cause a decline in snow cover and consequently in avalanche activity, while at higher elevation the increased precipitation is still mostly snow, leading to more, but due to higher temperatures, wetter snow cover, increasing (wet) avalanche activity.

To be able to study possible future trends of avalanche activity, future climate-model projections may be employed. The phases 3, 5, and 6 of the Coupled Model Intercomparison Project (CMIP; Meehl et al., 2007; Taylor et al., 2012; Eyring et al., 2016, respectively) now provide a large archive of future climate projections based on numerical global climate models (GCMs) run for the Special Report on Emission Scenarios (SRES) future emission scenarios (IPCC, 2000), the Representative Concentration Pathways (RPCs; Moss et al., 2010; van Vuuren et al., 2011), and the Shared Socioeconomic Pathways (SSP Riahi et al., 2017). However, the resolution of these GCMs on $\mathcal{O}(100)$ km is too coarse to represent meaningful data for robust projection of avalanche activity. Thus, to exploit the GCM outputs for purposes that require finer resolution, statistical and dynamical downscalings may be utilised. Due to the large computational costs of high-resolution dynamical downscalings, they are typically confined to specific regions, using the GCM simulation outputs as boundary conditions. For Norway, such a dynamical downscaling is provided in the form of the Nordic Convection Permitting Climate Projections (Lind et al., 2020, 2023), making available high-resolution (3 km) 21st-century climate projections for Scandinavia. In order to use the climate data for the investigation of future avalanche activity, statistical models may be invoked to find linkages between meteorological variables and avalanche occurrence. The statistical prediction of avalanche activity based on meteorological data has a long history and has recently gained momentum, likely spurred by advances in artificial intelligence technology (for reviews see Dkengne Sielenou et al., 2021; Eiselt and Graversen, 2025). The statistical models trained on historical meteorological data and avalanche records can then be applied to the GCM downscalings such as NorCP to project the avalanche activity into the future.

To the authors knowledge, the first study employing projected future development of avalanche activity or danger based on future emission scenarios was Lazar and Williams (2008), although Martin et al. (2001) appear to have been the first to investigate the change of avalanche activity under changing climatic conditions based on a statistical linkage between meteorological parameters and avalanches. By implementing constant positive perturbations of temperature and precipitation in their statistical model, they found for their study area in the French Alps that while new-snow avalanches declined, wet-snow





avalanches increased in frequency (at least relatively). Lazar and Williams (2008) employed the SRES scenarios B1, A1B, and

A1F1 (low, mid-range, and high emissions, respectively; IPCC, 2000) and a dynamical-statistical downscaling on the GCM outputs for the Aspen ski area in the Rocky Mountains to obtain the required high-resolution data for avalanche prediction. Additionally, they used a snowmelt runoff model as well as a snow mass balance model to derive more information on the snowpack and snow cover. By investigating changes in the timing of wet avalanche activity for the years 2030 and 2100, they found that already by 2030 wet avalanches occurred several days earlier than in the historical average (1980–2000). By 2100

the changes in timing depended strongly on the emission scenario, with wet avalanches occurring over a month earlier than in the historical average in the high-emission A1F1 scenario. A further important study of future avalanche activity change was conducted by Castebrunet et al. (2014), who used emission scenarios (B1, A1B, A2) similar to Lazar and Williams (2008), in a dynamical-statistical downscaling for the French Alps. Based on these data, they employed the French model chain SAFRAN-Crocus-MEPRA (Durand et al., 1999, 2009) to provide sophisticated snow and meteorological conditions to feed statistical

models to predict avalanche activity. The avalanche activity was investigated for the periods 2021-2050 and 2071-2100 and compared to the historical period 1961-1990. Similar to Lazar and Williams (2008), Castebrunet et al. (2014) found that wet avalanches tend to appear earlier in the season in the future climate. In general, they observed a decrease of the avalanche activity by 20–30 % throughout the 21st century compared to 1961-1990, both in the mean and in the interannual variability. However, there were differences depending on season (spring, winter) and elevation. The decrease in activity was pronounced

in spring and at low elevation, while winter avalanche activity in fact increased at high elevation. Latitude (at least across the study region of the French Alps) had only a small effect. Finally, Castebrunet et al. (2014) found only a relatively small dependence of the avalanche activity changes on the selected emission scenario, which, however, was still large enough that they noted that the current climate policies may have some effect on future avalanche activity. More recently, Katsuyama et al. (2023) studied weak layers in the snowpack in a +4 °C climate in Japan based on an ensemble of GCM projections using

a regional dynamical downscaling and the snow model SNOWPACK (Bartelt and Lehning, 2002; Lehning et al., 2002a, b; Morin et al., 2020). They found a general elevation-independent decrease of the probability of weak layer formation in the 4 °C warmer climate, mostly due to higher air temperatures, the decrease being only partly counteracted by an increase in snowfall intensity. However, as they pointed out, only dry avalanches were considered in their study, meaning that a potential wet avalanche increase as reported in some other studies could not be found by design. The most recent study on avalanche

activity change in the future climate is Mayer et al. (2024), expanding on Mayer et al. (2023a). They utilised the Swiss CH2018 climate change scenarios (CH2018, 2018; Fischer et al., 2022) based on the EURO-CORDEX dynamical downscalings (Jacob et al., 2014) for several RPCs (RPC2.6, RPC4.5, RPC8.5, implying low, medium, and high emissions, respectively) to project avalanche activity in the Swiss Alps throughout the 21st century. The data were further downscaled statistically to the locations of individual weather stations across the Swiss Alps. Both dry and wet avalanche activity were considered, based on the newly-

developed statistical models of Mayer et al. (2023b) and Hendrick et al. (2023), respectively. A clear decline of dry avalanche activity was found for all stations and scenarios, strongest (up to about 65 %) in RCP8.5 by the end of the 21st century. In contrast, the wet avalanche activity change was elevation dependent, with decreases below 2300 m a.s.l. and increases above. In total, above 2300 m a.s.l. the dry and wet activity changes mostly compensated each other (only about 10 % net decline).





For both wet and dry activity, differences between the RCPs were found, with avalanche activity changes generally levelling
off by mid-century in RCP2.6 and RCP4.5, while in RCP8.5 the decline was monotonic throughout the century. In an analysis
of the grain types simulated with SNOWPACK, Mayer et al. (2024) further indicated that the occurrence of persistent weak
layers associated with dry-snow avalanches also declines in the 21st century, consistent with the results of Katsuyama et al.
(2023).

The aim of the present study is to perform similar analyses of past and future avalanche activity change as presented in the
literature summarised above, however for the northern-Norwegian county of Troms. For this purpose we have the necessary
historical and climate projection data available from the 3 km Norwegian Reanalysis (NORA3) and the NorCP archive, respec-
tively. However, we first need to generate a suitable statistical model that can predict the avalanche activity or a related quantity
from these data. In Norway avalanche records are incomplete and the recording methodology has changed over time, likely
resulting in inhomogeneous data sets (Førland et al., 2007; Jaedicke et al., 2009). This is especially so in the sparsely populated
county of Troms. Figure 1, in addition to the above-mentioned fatalities, also depicts the number of observed snow avalanches
in Troms, which exhibits an exponential increase after the year 2000. Rather than an actual trend in avalanche occurrence, this
likely reflects the increasing usage of the database (Jaedicke et al., 2009), the establishment of the Norwegian Avalanche Warn-
ing Service (NAWS) in 2013 (Engeset, 2013; Müller et al., 2013), and the increasing winter recreational activity in Norway
(Engeset et al., 2018b), with people in the field reporting avalanches via online platforms (Engeset et al., 2018a). Given the
insufficient observational data, we here follow Eiselt and Graversen (2025) and train statistical models based on the information
from the Norwegian avalanche bulletin published daily on the online platform Varsom (www.varsom.no). In contrast to Eiselt
and Graversen (2025) we not only investigate the general avalanche danger level (ADL) but also the danger levels (DLs) of the
avalanche problems (APs) wind slab, persistent weak layer (PWL) slab, and wet snow. The DLs in the Norwegian avalanche
bulletin are in accord with the European Avalanche Warning Services' (EAWS) standard 5-level scale. To derive a measure
related to avalanche activity, we apply the binary classification from Eiselt and Graversen (2025) to differentiate avalanche days
(AvDs) and non-avalanche days (non-AvDs) based on the DLs. The number of AvDs per season is called the avalanche-day
frequency (ADF) and used as the metric to gauge the avalanche activity in the past and future northern-Norwegian climate.
Concurring with earlier work (e.g., Pérez-Guillén et al., 2022; Mayer et al., 2023b; Hendrick et al., 2023; Eiselt and Gra-
versen, 2025), random forest (RF) models (Breiman, 2001) are employed to predict the AvDs for the three APs and the general
avalanche danger, using predictive features generated from NORA3 and NorCP, including simulations with the physics-based
snow-cover model SNOWPACK to obtain more information on the snow stratigraphy.

The remainder of the study is structured as follows: Section 2 describes the data, comprised of the Norwegian avalanche
bulletin (2.1), the NORA3 reanalysis (2.2), the NorCP archive (2.3), the SNOWPACK simulations (2.4), and the predictive
features (2.5). Section 3, after briefly informing about the RF model (3.1) and the class balancing (3.2), explains the RF model
optimisation and feature selection procedure (3.3). Section 4 gives a brief analysis of the model performance and the feature
importances. The results are presented in section 5, first covering the past (5.1) and subsequently the future (5.2). Section 6
offers a discussion on the model performance (6.1), the past changes (6.2), and the future changes (6.3). Finally, section 7
summarises and concludes the study.





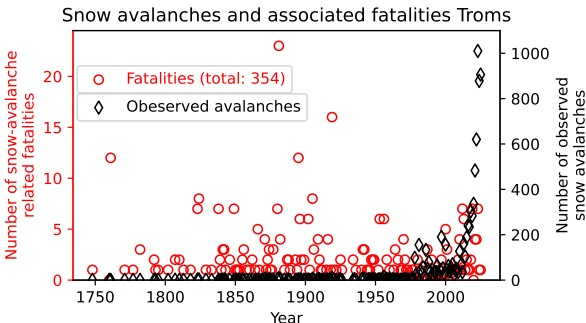

**Figure 1.** Time series of (red) snow-avalanche related fatalities and (black) observed snow avalanches in Troms, Norway. The data are provided by the Norwegian Water Resources and Energy Directorate (NVE) and can be downloaded from https://nedlasting.nve.no/gis/ (last access 29 January 2025).

## 2 Data

### 2.1 Norwegian avalanche bulletin

The Norwegian avalanche bulletin is published daily (before 16:00 LT) from 1 December to 31 May on the online platform Varsom (www.varsom.no; Johnsen, 2013; Engeset, 2013; Varsom, 2025) for the current day (nowcast) and the two following days (forecast). The main feature of the bulletin is the avalanche danger level (ADL) on the 5-level scale in accordance with the European Avalanche Warning Services' (EAWS) standards (EAWS, 2025a; Varsom, n.d.a) for 23 warning regions on mainland Norway (the average warning-region size is about 9000 km$^2$; Eiselt and Graversen, 2025). To determine the ADL, the forecasters first identify the active avalanche problems (APs), which are also published as part of the avalanche bulletin. In Norway, the following seven different APs are considered (Varsom, n.d.b), again following the EAWS' standards (EAWS, 2025b): new snow (loose and slab), wind slab, persistent weak layer (PWL) slab, wet (loose and slab), and gliding snow. For each of the identified problems a danger level (DL) is determined based on the estimated avalanche size and likelihood, the latter being derived from distribution and sensitivity (Müller et al., 2016, 2023). The final general ADL is then issued as the maximum of the DLs across the individual APs. However, this is not strictly applied and subject to the judgement of the forecasters, meaning that the final ADL does not always correspond to the maximum DL found across the APs.

Like Eiselt and Graversen (2025), we here consider the five northern-Norwegian warning regions of Nord-Troms, Lyngen, Tromsø, Sør-Troms, and Indre Troms (average size about 6800 km$^2$; see also Eiselt and Graversen, 2025) as shown in Fig. 2. These regions differ in terms of continentality (Fig. 2a; see also Dyrrdal et al., 2020) and elevation (Fig. 2b). Lyngen and especially Nord-Troms and Indre Troms exhibit much higher elevations than Tromsø and Sør-Troms. Accordingly, where relevant, the latter two will be referred to in the following as "low-elevation" regions and the former three as "high-elevation" regions.





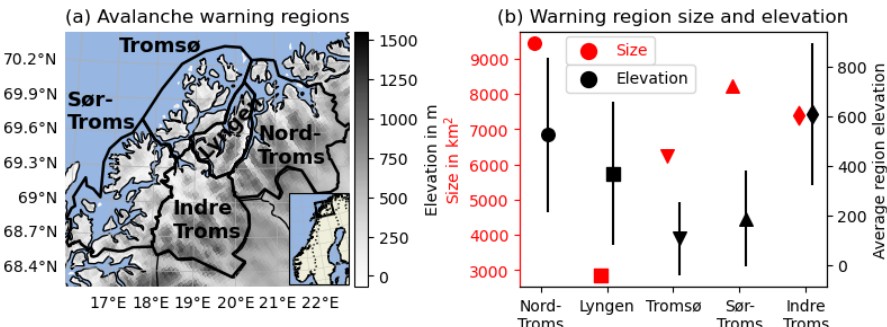

**Figure 2.** (a) Study region with NORA3 topography information and the borders of the individual warning regions (the figure is adapted from Fig. 1 in Eiselt and Graversen, 2025). (b) The size (red) of each warning region in $km^2$ and the average elevation (black) of each study region, including the standard deviation indicated by the error bars.

Instead of solely relying on the general ADL (as in Eiselt and Graversen, 2025), in this study the three APs wind slab, PWL
slab, and wet (loose and slab combined) snow are also considered. The Norwegian Avalanche Warning Service (NAWS) initiated the avalanche bulletin in 2013. However, since in the winter of 2016/17 the warning-region setup was changed (Karsten Müller, personal communication, 2024), continuous data availability encompasses the winters 2016/17 through 2024/25. For the present study, the general ADL was available during this entire period, but because of an issue regarding the reporting we could only obtain data from 2017/18 and onward for the individual APs. The nowcast data were downloaded from the Norwe-
gian Water Resources and Energy Directorate's (NVE) platform Regobs (https://www.regobs.no/, last access 5 August 2025, Engeset et al., 2018a) which is conveniently accessible with the Python library Regobslib (https://pypi.org/project/regobslib/, last access 5 August 2025). As shown in Fig. 3, the most frequently identified AP in northern Norway is the wind slab, followed by the PWL slab. The least frequent APs are new loose and slab snow and the glide slab; because of their infrequency they were not considered in this study. The general ADL averaged across all available values in the five considered warning regions
is about 2.2, being very similar to the average wind slab and wet snow DL. The PWL slab DL is on average somewhat higher at about 2.5, while the highest average DL is about 2.8 for the wet slab. However, the latter AP is only seldom identified and for the purposes of the current study was combined with the wet loose AP to obtain the wet snow AP.

The AP data downloaded from Regobs do not provide the DLs for the APs directly, but rather the parameters size, sensitivity and distribution. We use the methodology presented in Müller et al. (2016, 2023) to first convert sensitivity and distribution into
the likelihood and to subsequently determine the DL from likelihood and size (the procedure is available from Eiselt, 2025a).

To obtain a metric relating avalanche activity to the avalanche warnings, we follow Eiselt and Graversen (2025) and aggregate the DLs 1 and 2 to "non-avalanche days" (non-AvDs) and DLs $\geq 3$ to "avalanche days" (AvDs; see also Pérez-Guillén et al., 2024; Techel et al., 2024). The number of AvDs per season is referred to as the avalanche-day frequency (ADF) and is here interpreted as a metric that gives an indication of avalanche activity. The AvD/non-AvD aggregation was performed for the
general ADL as well as for the individual APs, wind slab, PWL slab, and wet snow.




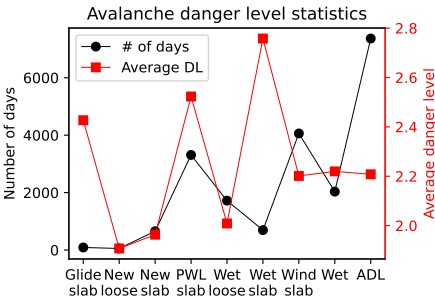

**Figure 3.** Avalanche problem (AP) frequency (black) and the average danger level per AP (red) in northern Norway on the days the specific AP was detected. The data cover the period from winter 2016/17 to 2024/25 for the general avalanche danger and 2017/18 to 2024/25 for the individual APs.

## 2.2 NORA3

As historical meteorological data, we utilise the 3 km Norwegian Reanalysis (NORA3). In fact, NORA3 appears as a mixture of a hindcast and a conventional reanalysis as it includes data assimilation only for surface parameters (Haakenstad et al., 2021; Haakenstad and Breivik, 2022). It provides a regional dynamical downscaling to 3 km horizontal resolution for northern Europe of the latest version of the European Centre for Medium-Range Weather Forecasts (ECMWF) reanalysis, ERA5, which has a 31 km horizontal resolution (Hersbach et al., 2020). To produce NORA3, the non-hydrostatic convection-permitting numerical weather model HARMONIE-AROME (Bengtsson et al., 2017) was run on a 3 km horizontal resolution and with 65 vertical layers, using ERA5 fields as initial and boundary conditions. At the time of writing, data availability covers the period from January 1970 to November 2024. Thus, our historical analysis comprises the winters 1970/71 through 2023/24. NORA3 is constantly updated with six to seven months lag, and an extension backward in time to 1960 is planned (Norwegian Meteorological Institute, n.d.b). For a more detailed summary on NORA3 see Eiselt and Graversen (2025).

## 2.3 NorCP

To investigate potential future changes in avalanche danger we exploit the Nordic Convection Permitting Climate Projections (NorCP; Lind et al., 2020, 2023). Similar to NORA3, NorCP provides a regional 3 km downscaling for northern Europe with a focus on Fenno-Scandinavia, but uses the output of two members of the Coupled Model Intercomparison Project Phase 5 (CMIP5) as boundary conditions. The simulations were performed with cycle 38 of the HARMONIE-Climate model (HCLIM38) on a 3 km horizontal grid using the AROME physics package, which is designed for models run at convection-permitting resolutions (Lind et al., 2020; for more details on HCLIM38 see Belušić et al., 2020). As noted by Lind et al. (2023), because of the large computational resource requirements of HCLIM38, no ensemble runs were performed, and instead the two CMIP5 members (CMIP6 had not yet been available at the start of the NorCP project) were chosen such as to represent a "middle-of-the-road" climate response with EC-Earth (Hazeleger et al., 2010, 2012) and a response with stronger warming and





precipitation increase with GFDL-CM3 (Griffies et al., 2011; Donner et al., 2011). For EC-Earth the two Representative Concentration Pathway (RCP) scenarios RCP4.5 and RCP8.5 (Moss et al., 2010; van Vuuren et al., 2011) were simulated, while, due to computational limitations, for GFDL-CM3 only RCP8.5 was run (Lind et al., 2023). The simulations were performed for a mid-century period (2040-2060) and a late-century period (2080-2100), in addition to a historical run (1985-2005). The RCP8.5 scenario has traditionally been referred to as the "business-as-usual" scenario, i.e. resembling a continuation of current global emission policies (Hausfather and Peters, 2020a). However, following recent developments, the Intergovernmental Panel on Climate Change (IPCC) acknowledged in their latest Assessment Report (AR6) that the likelihood of RCP8.5 should be considered low (IPCC, 2021, p. 238f.), as was also the original rationale (Moss et al., 2010). According to Hausfather and Peters (2020b), RCP4.5 should be seen as more likely, although this issue is not without debate (Schwalm et al., 2020).

In their evaluation of the NorCP model setup using ERA-Interim (Dee et al., 2011) as boundary conditions Lind et al. (2020) found that the downscaling especially improves the simulated precipitation with a more realistic representation of high-intensity events (this is similar to the improvments in NORA3; see Haakenstad et al., 2021; Haakenstad and Breivik, 2022). However, some important biases were also found, namely too-low near-surface temperature (mostly in summer) and too-strong winter precipitation, especially over complex orography. Lind et al. (2023) in their evaluation of the future projections found a pronounced elevation-dependent warming, the warming being up to 40 % larger at higher altitudes, specifically in the spring months, thus likely being important for avalanche danger in Norway. Moreover, they reported that days with significant snow depth quickly become less frequent over the 21st century, almost vanishing by the end of the century in the RCP8.5 simulations.

While the NORA3 and NorCP spatial resolutions are identical, the geographical coordinate systems are not. Thus, to ensure that the same grid-cell locations are selected for both datasets the NorCP data are re-gridded to the NORA3 coordinate system using the bilinear interpolation function *remapbil* from the Climate Data Operators (CDO) software (Schulzweida, 2023).

## 2.4 SNOWPACK simulations

To obtain more detailed information on the snow cover we run the physics-based, multi-layer model SNOWPACK (Bartelt and Lehning, 2002; Lehning et al., 2002a, b; Morin et al., 2020). The model solves the governing conservation equations (mass, energy, momentum) within the snowpack and simulates the snow cover one-dimensionally. For a summary of the key features of SNOWPACK see Morin et al. (2020).

Several approximations are implemented to derive the necessary input data for SNOWPACK from NORA3 and NorCP. From NORA3, the 2 m air temperature (TAS), relative humidity (RH), wind speed, wind direction, the net short-wave radiation at surface (NSW), and the precipitation amount are available as hourly values. For the ground temperature we make the simplification of using as a constant value the temporal mean of the TAS. For the surface temperature (TSS) we build a linear model from ERA5 data predicting TSS from TAS, wind speed, and long and short-wave net radiation at surface. For NorCP, instead of the NSW and TSS we use the incoming short- and long-wave radiation at surface. The RH in the NorCP data is 3 hourly and is here interpolated linearly to hourly values. The remaining parameters are the same as for NORA3.

To reduce computational resource requirements, and given the large warning regions, we perform a spatial aggregation of the 3 km NORA3 and NorCP data as follows: Four elevation bands (0-300 m a.s.l., 300-600 m, 600-900 m, and 900-1200





m) are defined, and for each elevation band, the data for the respective grid cells are averaged per warning region. These averages were then used as input for SNOWPACK, assuming flat terrain. This means that 20 SNOWPACK simulations were performed for the NORA3 historical data as well as for each NorCP future climate scenario. For the training and test data we additionally performed the SNOWPACK simulations for the four main aspects at 38° steepness (the most frequent avalanche

slope according to McClung and Schaerer, 2006), but the inclusion of these data did not improve the performance of the machine-learning models in predicting avalanche danger. Thus, we decided to continue our analysis only with the flat terrain data. In further tests, we attempted selecting those individual grid cells per warning region that are specifically exposed to wind and snow (based on a suggestion from Dyrrdal et al., 2020). SNOWPACK was then run for ten of these grid cells per warning region, but this also did not improve the machine-learning model performance. Thus, we use the spatially aggregated data as

explained above for our analysis.

## 2.5  Predictive features for avalanche danger

The predictive features used as input for the machine-learning models are presented in Tables B1 and B2 in Appendix B. They are selected based on basic physical understanding and on the results from earlier work (Zeidler and Jamieson, 2004; Mitterer and Schweizer, 2013; Conlan and Jamieson, 2016; Pérez-Guillén et al., 2022; Hendrick et al., 2023; Eiselt and Graversen,

2025). The meteorological features are similar to Eiselt and Graversen (2025), but important differences exist in terms of the snow-cover information. Here the complex snow model SNOWPACK is utilised instead of the simple model seNorge (Saloranta, 2012, 2014, 2016). The predictive features derived from the SNOWPACK output include the snow depth (SD) as well as several stability indices (see Table B2). To derive these stability indices, the threshold sum approach (TSA; Monti et al., 2012) with the thresholds given in Monti et al. (2014) was performed on the SNOWPACK output to find the weak layers in

the snowpack. Following Pérez-Guillén et al. (2022), we then determine one to two weak layers to extract the SNOWPACK-calculated stability indices Sk38 (skier stability index, Föhn, 1987; Jamieson and Johnston, 1998; Monti et al., 2016), Sn38 (natural stability index, Föhn, 1987; Jamieson and Johnston, 1998; Monti et al., 2016), and the structural stability index (SSI; Schweizer et al., 2006). If a weak layer is found within the first 100 cm of the snowpack the suffix _100 is appended to the index name and the search is continued to find a deeper weak layer. The indices from the deeper layer (if found) are denoted

with the suffix _2. Generally, larger values of these indices indicate a more stable snowpack.

A further important difference compared to Eiselt and Graversen (2025) is the spatial aggregation of the features. They tested averages and percentiles over different elevation bands and found that this only had a small effect on the results. We here attempt to integrate more information in the predictive features. After averaging the NORA3 data for each individual warning region separately over the four elevation bands 0-300 m a.s.l., 300-600 m, 600-900 m, and 900-1200 m, we take the maximum

and/or minimum of the features (depending on the feature; this is indicated with the suffixes _emin and _emax, respectively) over all elevation bands. We have tested generating the predictors for ten specifically wind- and snow-exposed grid cells per avalanche region (SNOWPACK was run for these grid cells specifically as well, see section 2.4), but this did not improve the performance of the machine-learning models. Hence, we base our analysis on the spatially aggregated predictive features.





# 3 Methods

## 3.1 Machine learning – random forest

To establish the statistical linkage between meteorological data and avalanche danger we employ the widely used random forest (RF) model (Breiman, 2001). We use and further develop the methodology Eiselt and Graversen (2025, see Eiselt, 2024) using the RF implementation from the Python library scikit-learn version 1.3.0 (https://scikit-learn.org/, last access 23 September 2025). The reader is referred to Eiselt and Graversen (2025) for more details.

## 3.2 Class balancing – synthetic minority over-sampling

To account for the imbalanced class frequencies in our data, which may lead to biased model training, we follow Eiselt and Graversen (2025) in using the synthetic minority over-sampling technique (SMOTE; Chawla et al., 2002; Fernandéz et al., 2018) to oversample the minority class. The SMOTE algorithm generates new samples of the minority class by interpolating between existing samples. Like Eiselt and Graversen (2025) we use the implementation of the SMOTE algorithm in the Python library imbalanced-learn version 0.12.3 (https://imbalanced-learn.org/, last access 23 September 2025).

## 3.3 Random forest optimisation and feature selection

When investigating different combinations of hyperparameters for the RF model, we find a large variation of model skill. This variation mostly derives from the hyperparmeters min_samples_leaf (MSL) and min_samples_split (MSS), while the other hyperparameters (max_depth, n_estimators, max_features) appear to have a much smaller influence. Given these hyperparameter dependencies we deviate from the model optimisation and feature selection procedure conducted in earlier work (e.g. Pérez-Guillén et al., 2022; Hendrick et al., 2023; Eiselt and Graversen, 2025). That is, we do not perform a randomised grid search over all the different hyperparameters, but instead only test different values of MSL and MSS.

Following the notion of Winkler and Murphy (1992) that "the search for a single 'best' measure is doomed to failure," the decision is here made to not focus solely on one performance measure. Instead, we consider such measures as the accuracy or percentage correct (PC), the false alarm rate (FAR), the F1-macro score, and the true skill score (TSS). The metrics are presented in Appendix C. While using several metrics to optimise the model may introduce some inconsistency into the procedure, it lowers the danger of optimising a single performance metric at the expense of others. For instance, optimising the model to a high PC may come at the expense of an unacceptably high FAR.

The procedure of hyperparameter and feature selection applied here works in the following five steps:

1. A leave-one-out validation is performed across the training data using all available predictive features (see Tables B1 and B2) to find the best MSL and MSS hyperparameters (see Figs. S1 to S4 in the Supplement). That is, since five years of training data are available, the model is trained on four years, while one year is excluded and used for validation. This is repeated for each year and we here consider the average of the above-listed performance metrics over the excluded validation years. Note that we average over the years but not over the performance metrics. As mentioned above, we have



repeated this procedure with different values of other hyperparameters (max_depth and n_estimators), but the impact on the results was marginal.

2. An initial RF model is trained with the hyperparameter combination from step one. This gives a best-features ranking based on feature importances which is then used to perform an iterative feature search, where those features are excluded that exhibit a correlation of R > 0.9 (Pearson R) with a feature of higher importance.

3. The first step is repeated with the reduced set of best predictive features found in the second step.

4. The RF model must be trained again with the hyperparameter set found in step three and the list of features from step two to produce a new feature-importance ranking.

5. The ranking from step four is use to find the best number of predictive features by again investigating the performance metrics mentioned above for different numbers of features.

We optimise and train four different RF models, each for one specific AP including the general ADL. This is necessary as the different APs are linked to different weather and snowpack conditions. More specifically, earlier research on statistical prediction of avalanches has shown that different predictive features are important for the different APs (compare the different most important features found in Pérez-Guillén et al., 2022; Hendrick et al., 2023; Zeidler and Jamieson, 2004).

## 4   Model performance and feature importances

Figure 4 shows the confusion matrices for the four RF models trained for the four different APs. It is clear that the model skill depends on the AP. However, in general, all RF models are better at predicting non-AvDs than AvDs. Considering both non-AvD and AvD, the best and most balanced performances are achieved for the wind slab AP and the general avalanche danger with a recall score (RC; see Appendix C) between 0.66 and 0.83. For the PWL slab and especially for the wet AP the non-AvDs are also mostly correctly predicted (RC=0.81 and RC=0.90, respectively). However, while for the wet AP the AvD
prediction works reasonably well (RC=0.60), for the PWL slab AP the performance is weak (RC=0.35).

Figure 5 shows the feature importances of the 15 most important predictive features for each of the APs. The numbers of included features in the RF models after the optimisation procedure are presented in Table D1 in Appendix D. In general, the different APs have different most important features, although the wind slab and the general problem exhibit similarities. For the wind slab the most important parameters are related to new snow (s7_emin, s3_emin), snow-drift (wdrift3_3_emax), and
wind speed (wmax3_emax). However, some SNOWPACK-derived parameters are also important, such as the 1 to 3 d change of the Sn38 stability index (Sn38_100_d1...3). For the PWL slab the change over time of the SNOWPACK-derived structural stability index (SSI_2_d3, SSI_100_d3) is most important, while for the wet AP the maximum temperature (tmax_emax) has the highest importance. As mentioned above, the general avalanche danger is similar to the wind slab AP in terms of most important features with new snow and wind-related features being most important. However, the SNOWPACK-derived





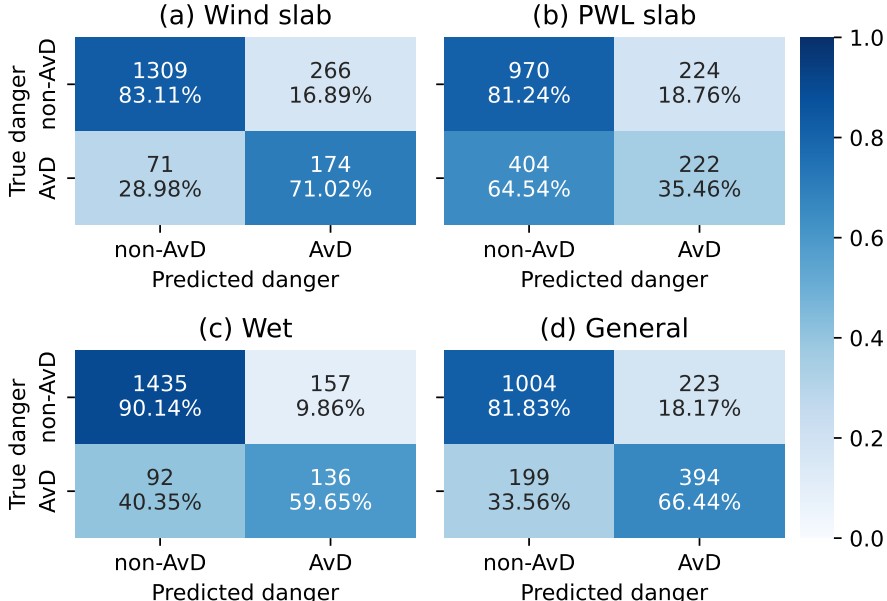

**Figure 4.** Confusion matrices for the random forest models trained for the different avalanche problems: (a) wind slab, (b) PWL slab, (c) wet, and (d) general. The abbreviations AvD and non-AvD mean avalanche day and non-avalanche day. The colour indicates the fraction of the number of instances. Note that in the matrices the upper-left and lower-right cells correspond to the recall score (see eq. C3 in Appendix C) for the non-AvDs and AvDs, respectively.

stability indices appear less important for the general problem and instead the seven-day averaged net short-wave radiation (nsw7_emax) plays a role as well as the change of the LWC (lwc_i_d2).

## 5   Results

### 5.1   Past changes – NORA3

Like Eiselt and Graversen (2025) we perform a hindcast of the ADF for the NORA3 period from 1970 to 2024. The main
differences to Eiselt and Graversen (2025) are (1) that here the predictive features include the SNOWPACK output and (2) that the predicted avalanche danger is differentiated into individual avalanche problems. The annual and 7 y rolling mean hindcasts for the wind slab, PWL slab, and wet problems, as well as the general ADF for the warning region Lyngen during winter (Dec–Feb) are shown in Fig. 6 (Fig. S5 in the Supplement shows the general ADF in Lyngen for spring, winter, and the full season).





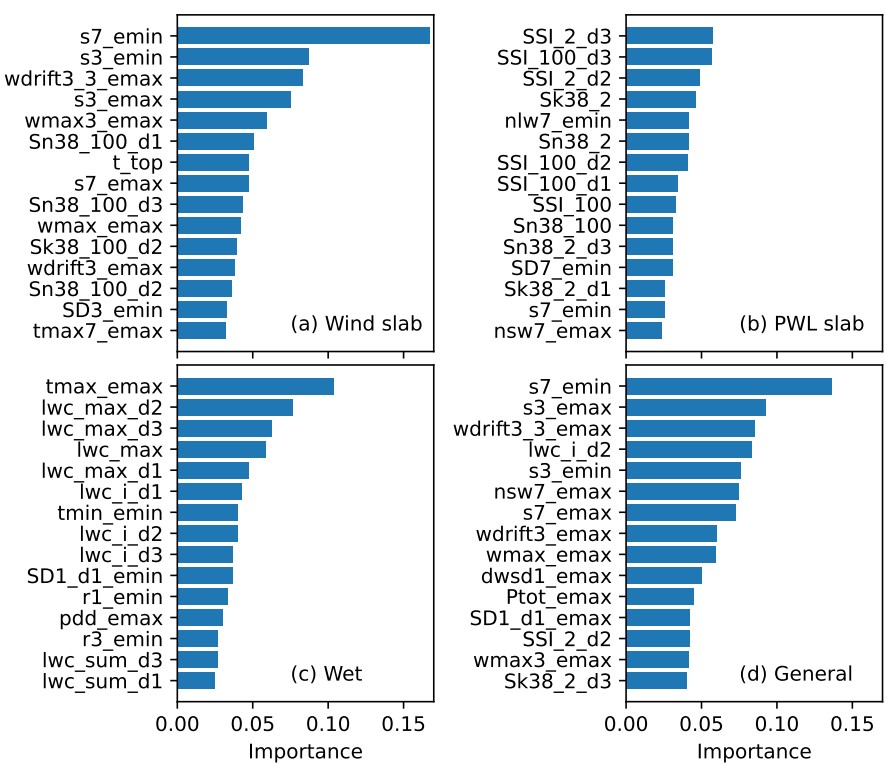

**Figure 5.** The fifteen most important features and their feature importances for the random forest models for the different avalanche problems: (a) wind slab, (b) PWL slab, (c) wet, and (d) general. See Tables B1 and B2 in Appendix B for descriptions of the predictive features.

### 5.1.1 Linear trends

We first consider linear trends of the ADF over the whole NORA3 period (1970–2024). The values of the trend slopes are depicted in Fig. 7 (see also the straight lines in Fig. 6 for Lyngen) and the regression coefficients (Pearson R) are shown in Table S1 in the Supplement.

The main result is that overall, few significant linear trends are observed. However, a general aspect is that trends in winter (Dec–Feb) are negative, while the spring (Mar–May) trends are positive, resulting in marginal trends throughout the full avalanche season (Dec–May). This is consistent across warning regions and APs (except for the wet problem). For the wind slab problem the full season trends are negative, but for all other problems they are mostly positive.

### 5.1.2 ADF–AO linkage

Prompted by the significant correlations of the winter ADF with the Arctic Oscillation (AO) index found by Eiselt and Graversen (2025) we here also investigate this linkage. Figure 8 shows the correlation coefficients of the AO index with the ADF for all regions, seasons, and APs both for the annual and 7 y rolling mean values (see also Table S2 in the Supplement).





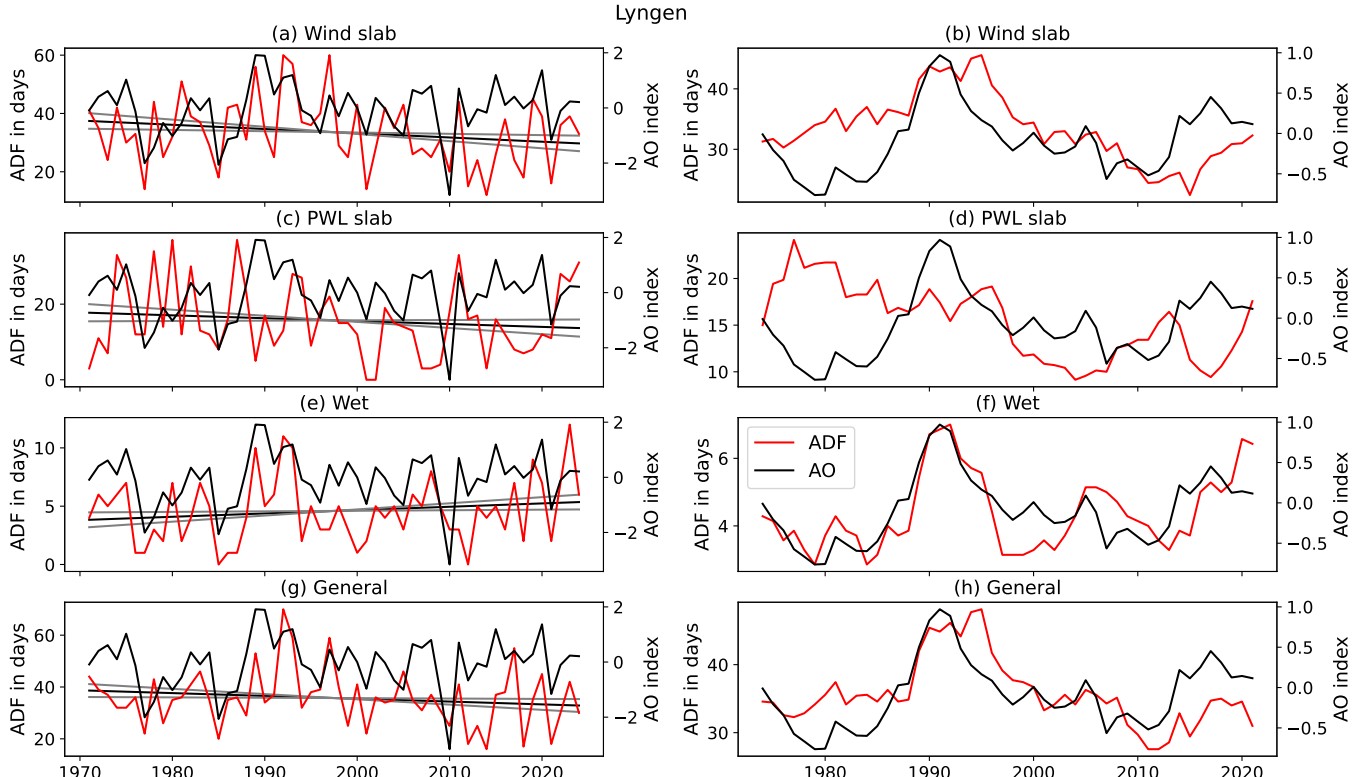

**Figure 6.** Winter (Dec-Feb) avalanche day frequency (ADF; red) and Arctic Oscillation (AO) index (black) from 1970 to 2024 based on NORA3 data for (a, b) wind slab, (c, d) PWL slab, (e, f) wet, and (g, h) general avalanche problem for (left panels) annual and (right panels) 7 y rolling means in the warning region Lyngen. The straight black and gray lines indicate the linear trends and their uncertainties. Note the different y-axis scales for the different avalanche problems. For the corresponding correlation coefficients of AO and ADF see Fig. 8 and Table S2 in the Supplement.

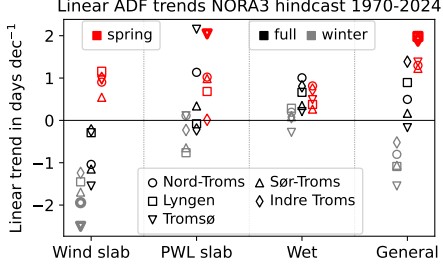

**Figure 7.** Linear trends in the full season (Dec-May), winter (Dec-Feb), and spring (Mar-May) avalanche day frequency (ADF) from 1970 to 2024 based on NORA3 data for all avalanche problems and regions. Bold markers indicate statistical significane at p < 0.05.





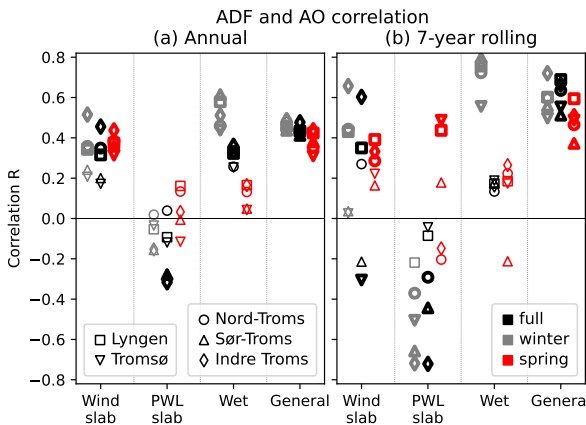

**Figure 8.** Correlation coefficients (Pearson R) between avalanche day frequency (ADF) and Arctic Oscillation (AO) index for all seasons, regions, and avalanche problems. Bold markers indicate statistical significance at p < 0.05. The values are also shown in Table S2 in the Supplement.

The main result is that there is a significant linkage between the AO and the ADF during winter (Dec–Feb) for all APs, except the PWL slab. The results for the individual seasons and APs are presented below.

It is clear from Fig. 8 that the most consistent correlations across regions and seasons are found for the general ADF. As in
Eiselt and Graversen (2025), correlations are stronger for the 7 y rolling means (0.37–0.72) than for the annual values (0.31–0.49), but they also vary more than for the annual values across regions, and correlations tend to be stronger in winter than in spring. The correlations of the AO index and the wind slab ADF are similar to the general ADF, but slightly weaker and more variable across regions and seasons. The weakest correlations of AO and ADF are found for the PWL slab problem for the annual values which exhibit mostly correlations close to zero. In contrast, for the 7 y rolling means, some strong and significant
negative correlations are found in the winter and the full season. However, these vary strongly across regions (-0.72−-0.04), and in spring even some significant positive correlations are evident. The strongest correlations of the AO index and the ADF are observed for the wet AP in winter, both for the annual (0.44–0.61) and 7 y rolling means (0.56–0.80). However, in spring, the wet ADF does not appear to correlate with the AO index. We note that the wet ADF with values of about 5 d is considerably smaller than the ADF of the other problems (20–40 d). As an example, the ADF values for all APs are shown for Lyngen in
Fig. 6 for winter together with the winter AO index.

## 5.2 Future changes – NorCP

To investigate potential future changes of avalanche danger and problems, we consider the NorCP downscalings based on the EC-Earth climate model for the historical as well as RCP4.5 and RCP8.5 scenarios (Fig. 9). We find that results based on the NorCP downscaling of the GFDL-CM3 RCP8.5 scenario are generally similar to the corresponding EC-Earth-based
simulations, as can be seen in Fig. S6 in the Supplement. The significance of the changes of the ADF between historical,



mid-century, and late century is tested with a Monte-Carlo simulation. The numerical changes and their statistical significance are shown in Figs. S7 to S10 in the Supplement.

The main result is that the ADF in northern Norway declines over the 21st century for all APs, except for the wet snow AP, which in fact exhibits an increase for the high-elevation regions (Nord-Troms, Lyngen, Indre Troms). We present the results in

more detail below, focusing first on RCP4.5, followed by RCP8.5.

The wind slab ADF in RCP4.5 declines monotonically in all regions. The changes are similar across the regions and statistically significant an all cases ($p < 0.05$; see Fig. S7).The ADF development of the PWL slab AP is less clear. A slight decline in the early period can be observed, but this is significant only in Nord-Troms and Indre Troms (Fig. S8). In the late period the changes are similar, implying a statistically significant decline of the PWL slab ADF from the historical to the

late-century period. The wet problem differs from all other APs in that there is a general increase in ADF. During the early period, the increase is statistically significant and consistent across all regions (Fig. S9). In the late period only Tromsø exhibits a significant decline. Throughout the whole century, the high-elevation regions (Nord-Troms, Lyngen, Indre Troms) show a significant increase in wet ADF, while in the low-elevation regions (Tromsø, Sør-Troms) there is no significant change. For the general avalanche danger all regions show a decline throughout the century. However, in the early period the changes are only

significant in Tromsø and the decline in Indre Troms never obtains statistical significance (Fig. S10).

In RCP8.5 the wind slab ADF changes are generally similar to RCP4.5 although the decline is more severe and tends to accelerate in the late century. All changes (throughout the century and in the individual early and late periods for all regions) are statistically significant (Fig. S7). Notably, in the low-elevation regions of Tromsø and Sør-Troms the late-century ADF is close to zero, while in the other three (high-elevation) regions the ADF, although low, is different from zero. However,

the difference is statistically significant ($p < 0.05$) only in Indre Troms (based on a Monte-Carlo simulation). The changes in the PWL slab ADF in RCP8.5 during the early period appear similar to RCP4.5, meaning a small, often non-significant decline. However, in the late period the decline accelerates in all regions and becomes statistically significant. Again, as for the wind slab AP, in the late century the low-elevation regions PWL slab ADF is almost zero, while the other regions retain non-zero ADF (Fig. S10). Again, this is statistically significant only for Indre Troms. The RCP8.5 wet ADF shows a rather

dichotomous behaviour between low- and high-elevation regions: After a significant increase that is similar in all regions in the early period, the increase continues in the high-elevation regions, while in the low-elevation regions a decline sets in. Notably, in the high-elevation regions this leads to the wet problem being the dominant AP in the late century, while historically and in the mid-century the wind slab and the PWL slab were more dominant. Conversely, in the low-elevation regions, the wet ADF declines to a level in the late century that is statistically significantly smaller than the historical level (Fig. S9). Finally, the

general ADF in RCP8.5 again declines significantly throughout the century (Fig. S10), with a slight acceleration in the late period in all regions.




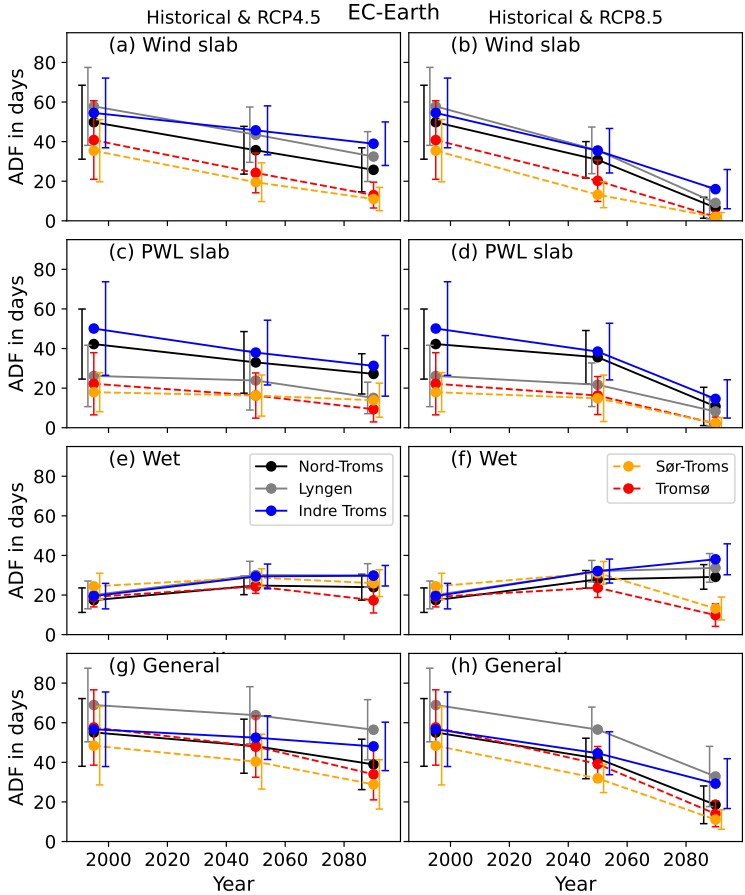

**Figure 9.** Full-season avalanche day frequency (ADF) averaged separately for the three 21-year NorCP periods (historical, mid-century, and late century) for (a, b) wind slab, (c, d) PWL slab, (e, f) wet, and (g, h) general avalanche problem. The different colours represent the different warning regions. The low-elevation regions are depicted with dashed lines, the high-elevation regions with solid lines. The errorbars indicate the 1-$\sigma$ standard deviation. The errorbars are slightly horizontally shifted for better readability. The left panels show the historical and RCP4.5 and the right panels the historical and RCP8.5 simulations with EC-Earth. For the historical and RCP8.5 simulations with GFDL-CM3 see Fig. S6 in the Supplement. The Figs. S7 to S10 in the Supplement show the numerical changes between periods and indicate the significance of the changes.

# 6 Discussion

## 6.1 Model skill and feature importances

As shown in section 4, the model performance varies considerably between the different APs, especially with respect to the prediction of AvDs, while for non-AvDs the performance is good in all cases. A potential reason for this is the much higher frequency of non-AvDs compared to AvDs (Fig. 4), i.e. an apparently high model skill can be achieved by simply defaulting to




non-AvD. However, for the PWL slab 34 % of days are AvDs as compared to only 13 % for the wind slab and wet APs, while the model performance for the latter problems is considerably better than for the former. This indicates that the prediction of the PWL slab AP is fundamentally more difficult than the prediction of the other APs. Notably, while the wind slab AP mostly depends on the current weather conditions, the PWL slab additionally depends on the conditions earlier in the season (e.g., Marienthal et al., 2015). The earlier weather conditions are included in our predictive features in so far as they impact the SNOWPACK simulations. However, this information is rather uncertain due to its heavy model-based character, and the large-scale spatial aggregation performed in the preparation of the predictive features (see sections 2.5 and 2.4). Thus, it appears plausible that in our setup the wind slab is more predictable than the PWL slab. The wet AP is to a large degree determined by the maximum temperature, implying a stronger predictability compared to the PWL slab. The weaker performance of the wet AP prediction compared to the wind slab may reflect the lack of understanding of the mechanics of wet-snow avalanches as noted by Mitterer et al. (2013). The survey among forecasters conducted by Techel and Pielmeier (2009) emphasises that the timing of wet-snow avalanches is a major difficulty, likely contributing to less accuracy when forecasting the wet AP.

That the most important features of the wind slab and general APs are mostly related to wind, new snow, and snow depth is consistent with earlier work covering Norway (Bakkehøi, 1987; Kronholm et al., 2006; Jaedicke et al., 2008; Eiselt and Graversen, 2025) and other regions (Davis et al., 1999; Gauthier et al., 2017; Pérez-Guillén et al., 2022; Hao et al., 2023) and is extensively discussed in these studies. The 1 to 3 d changes of the Sn38 and Sk38 stability indices also appear important for the wind slab, although in earlier work these changes are more typically associated with PWL slab avalanches (e.g. Zeidler and Jamieson, 2004). However, note that here these features relate only to shallow weak layers (Sk38_100, Sn38_100), potentially indicating more immediate avalanche release. Indeed, as seen in Fig. 5b for the PWL slab AP, the stability indices at deeper weak layers (SSI_2, Sk38_2 etc.) are generally more important than at the shallower weak layers. This consistency indicates that despite the above-mentioned high uncertainty of the SNOWPACK-derived features, these still are of some value in predicting avalanche danger (see also Schirmer et al., 2009). While we here find the most important features for the PWL slab AP to be related to SNOWPACK-derived stability indices (consistent with Zeidler and Jamieson, 2004), other studies have emphasised the importance of meteorological parameters. For example, Marienthal et al. (2015) and Conlan and Jamieson (2016) noted the impact of air temperature and its changes for PWL slab hazard. While these do not appear explicitly among the most important features for the PWL slab here (Fig. 5b), they may still be indirectly represented via the stability indices due to their impact on the snowpack. Furthermore, earlier work pointed to the importance of snow loading (Marienthal et al., 2015; Conlan and Jamieson, 2016), which appears incorporated here as well by the new snow and snow depth parameters (Fig. 5b). Finally, Conlan and Jamieson (2016) showed the relevance of solar warming for the PWL slab hazard. Consistently, the short-wave radiation (nsw7_emax) is also among the features determining the PWL slab here. However, while Conlan and Jamieson (2016) emphasised that in addition to the spring the solar radiation is also important in the winter months, this cannot be the case here, since polar-night conditions prevail for most of the northern Norwegian winter. Thus, there is no short-wave radiation during the winter months and its influence must appear in the spring months.

Regarding the wet problem our results appear consistent with earlier work showing the importance of air temperature and liquid water content (LWC) of the snow (Mitterer and Schweizer, 2013; Hendrick et al., 2023). Specifically, similar to Hendrick





et al. (2023) for the LWC features and the snow depth, mostly the temporal 1 to 3 d changes are important. In addition, the 1 to 3 d changes of the LWC index (lwc_i) introduced by Mitterer and Schweizer (2013) appear among the most important features of the wet AP, supporting the applicability of this index.

## 6.2 Past changes – linear trends and correlation with the Arctic Oscillation

### 6.2.1 Linear trends

The lack of significant long-term (1970-2024) trends of the ADF found in section 5.1.1 is consistent with the findings of Eiselt and Graversen (2025), based on similar data, but also with Saloranta et al. (2024) who found only few significant trends in their analysis of avalanche indicators in Norway for the period 1961 to 2020. Dyrrdal et al. (2020) investigated trends in climate indices connected with avalanche activity for Troms in northern Norway over the period 1958 to 2017 for October through April. Their results were also somewhat mixed, with variables related to snow (e.g. maximum snow amount) exhibiting both positive and negative trends across Troms, while winter rain events consistently increased. This appears consistent with our results as the wet ADF and in some regions the general ADF exhibit positive trends from 1970 to 2024 (Dec–May; Fig. 7). However, it must be mentioned again that these trends are non-significant and our study appears in line with the statement from the review of Eckert et al. (2024) that "trends in the the number or frequency of avalanches are often elusive" due to avalanche cycles and large decadal variability. In northern Norway, the avalanche activity appears to be influenced by regional climate modes (Eiselt and Graversen, 2025), which we turn to in the next section.

### 6.2.2 ADF–AO linkage

The linkage between the general ADF and the Arctic Oscillation(AO) was explored and extensively discussed by Eiselt and Graversen (2025). Several earlier studies also indicated a likely influence of the AO-related North Atlantic Oscillation (NAO) on northern Norwegian avalanche activity through its impact on the meteorological conditions (e.g. Uvo, 2003; Dyrrdal et al., 2020), although the explicit correlation analysis of the AO/NAO with ADF in Norway had not previously been conducted (for other countries such as, e.g., Iceland see Keylock, 2003). Eiselt and Graversen (2025) found especially the 3 d sum of new snow as well as the 3 d averaged wind speed in northern Norway to be correlated with the AO index. As these parameters in their combination constituting snow drift (wdrift3_3) were most important in determining the ADF, this explained the causal linkage of the ADF with the AO. Here we also investigate the linkage of the predictive features with the AO and find similar strong correlations of the accumulated new snow and wind speed with the AO index (Fig. 10). However, because of the aggregation of the predictive features across the different elevation levels as maxima ("emax" features) and minima ("emin" features; see section 2.5), which is different from Eiselt and Graversen (2025), there is additional complexity to our results. Indeed, we find that the emin features (s3_emin, s7_emin) are only weakly correlated with the AO index, while the correlation with the emax features (s3_emax, s7_emax) is strong (Fig. 10). Thus, it appears that the AO partly determines the maximum snow fall and depth across elevations in northern Norway but has little effect on the minimum. This is likely also the reason for the weaker correlation of the AO index with the wind slab ADF than with the general ADF (Fig. 8), despite their similar most important





features (compare Fig. 5a and d). For the general ADF the second most important feature is an emax variable (s3_emax) while
for the wind slab both first and second most important are emin variables (s7_emin, s3_emin). An additional contributing
factor is that s7_emin is relatively more important for the wind slab ADF than for the general ADF (compare Fig. 5a and d).
Nevertheless, several of the features determining both the general and the wind slab ADF are correlated well with the AO index
(Fig. 10), explaining the considerable linkage of both APs with the AO. Eiselt and Graversen (2025) focused mostly on the
winter months December through February, for which the linkage was found to be clearest, although there are also significant
correlations of the AO index with the general and wind slab ADFs in the spring months March through May (Fig. 8). However,
here we point out that the character of the correlation in spring is different from that in the winter. The conspicuous peak of the
general ADF visible both in the winter (for Lyngen see Fig. 6) and the full season is absent in the spring (see for Lyngen Fig.
S5 in the Supplement). Instead, subsequent to little change from 1970 to 2005, a continuous increase in the ADF is apparent,
concurrent with an increase in the AO index resulting in the significant correlations seen in Fig. 8. The most important feature of
both the general and the wind slab ADF (s7_emin) exhibits a corresponding rise at the same time (not shown), likely explaining
the ADF development. Given the weak correspondence with the AO index in the 1990s it is questionable if the increase of the
spring ADF is caused by the AO. Rather, it appears that the generally increased precipitation due to Arctic warming (see also
Dyrrdal et al., 2020) is the main driver of the rise in general and wind slab ADF in spring.

Expanding the discussion to the other APs, we find that the strong correlations of the wet ADF with the AO index in winter
(Dec–Feb) are also well explained by the impact of the AO on the most important meteorological features for this AP. Both
tmax_emax and the features related to the LWC (e.g. lwc_max_d2 and lwc_max) are strongly correlated with the AO index in
winter (Fig. 10). Conversely, correlations of the wet ADF with the AO index in spring (Mar–May) are much lower, especially
for the annual values. This is well in line with the most important feature (tmax_emax) which shows only a small peak in
the 1990s, and after 2005 remains constant or even slightly declines, while the spring AO index concurrently increases (Fig.
S11 in the Supplement). Spring temperatures thus appear little influenced by the AO explaining its small impact on the spring
wet ADF. The low correlations of the full-season (Dec–May) wet ADF with the AO index are more difficult to understand
since most of the important features (e.g. tmax_emax and lwc_max_d1) are correlated well with the AO index (Fig. 10). One
explanation may be that for several other important features (lwc_i_d1, SD_d1_emin), the correlations with the AO index are
considerably worse for the full season than winter.

Finally, a potential linkage of the PWL slab ADF with the AO appears unclear, especially since there are large differences
between the annual and 7 y rolling-mean values. For the annual values the correlation of the PWL slab ADF with the AO index
is mostly close to zero, which is consistent with correlation between the AO index and the most important feature (SSI_2_d3;
compare Figs. 8 and 10). For the 7 y rolling mean values the in some instances strong negative correlations of winter and
full-season PWL slab ADF with the AO index also appear in line with the correlations between the AO index and SSI_2_d3.
We are unable to provide an explanation for the differences between the annual and the 7 y rolling mean values. However
two points may be noted: (1) The generally negative correlation of the PWL slab ADF with the AO index may result from
the concurrent higher wind slab and wet ADF, reflecting the fact that fewer weak layers persist for a long time as avalanches





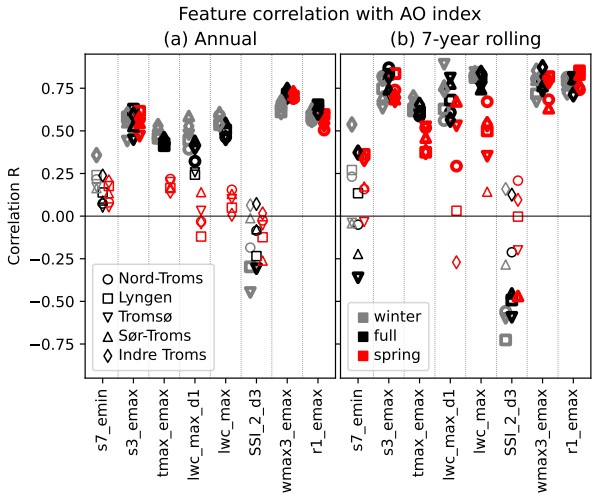

**Figure 10.** Correlation (Pearson R) of NORA3- and SNOWPACK-derived predictive features with the Arctic Oscillation (AO) index for all regions in winter (gray), spring (red), and full season (black). Bold markers indicate significant correlations at p < 0.05. For the feature definitions see section 2.5 and Tables B1 and B2 in Appendix B.

readily release due to frequent new snow and wind-drift loading. (2) The performance of the RF model in terms of predicting PLW slab ADF is low (Fig. 4), calling into question the robustness of the results regarding this AP.

## 6.3 Future changes – dependence on avalanche problem and elevation

Similar to the previous section for the historical ADF development, we investigate projected changes of the most important predictive features to explain the changes of the ADF in the future climate scenarios.

The wet problem stands out among the APs as it is the only one to mostly increase over the 21st century, dependent on the warning region. The increase appears well in line with the important features (Fig. 5), i.e. the maximum temperature (tmax_emax; see Fig. 11e, f) and several LWC-derived features (lwc_max_d2, lwc_max; see Fig. 11g, h and Fig. S12e, f, respectively) which increase as well. The dependence on the warning region is likely explained by the changes in snow depth across different elevation levels: In the low-elevation regions (Tromsø and Sør-Troms) there remains little snow in the late century (see also Dyrrdal et al., 2020), leading to a general decline in all the APs, including the wet problem, while in the other regions there is still enough snow to cause considerable avalanche danger. We note that when not including the snow depth (or a derived parameter) as a predictive feature in our RF model we found that the model predicted a similar increase in wet ADF for all warning regions even in the RCP8.5 late-century simulation. This is likely due to the temperature being the most important feature, determining a continuous increase in wet ADF. However, this becomes unphysical if no snow remains. Thus, care should be taken to include snow depth or snow-depth related parameters as predictive features in machine-learning models, even if they do not appear among the most important (see also Lazar and Williams, 2008, and Appendix A).





Turning to the other APs, the consistent decline of the wind slab and general ADF throughout the 21st century results partly from a significant decline of the snow fall (s7_emin; see Fig. 11a, b) as well as other important parameters related to snow depth, wind speed, and snow drift (see Fig. S12 in the Supplement). This is in agreement with earlier work from Dyrrdal et al. (2020) who investigated RCP8.5 future projections of climate indices related to avalanches in Troms. They found a decline in most of the indices related to snow fall and wind (maximum snow fall, snow drift), especially towards the end of the century

and indicated that a decline of dry-snow avalanches may set in before 2040, even in the higher and colder regions within Troms, consistent with the results presented here.

The decreasing PWL slab ADF over the 21st century concurs with the changes in the stability indices and their derivative features. The 3 d change of the SSI (SSI_2_d3) becomes smaller (Fig. 11c, d) while the SSI itself increases (Fig. S12a, b), implying more stable snowpack conditions. The decline of the PWL slab ADF is consistent with the recent study for the

Swiss Alps by Mayer et al. (2024). They conducted a more detailed analysis of the SNOWPACK profiles and found a decline of persistent grain types, which tend to form in cold conditions, decreasing the likelihood of PWL formation. They further pointed to the influence of higher temperatures being responsible for weaker temperature gradients through the snowpack, leading to fewer weak layers being formed. We note that although temperature-related features are not important in the prediction of the PWL slab AP (Fig. 5b), the temperature effect is represented in the SNOWPACK-derived stability indices (SSI, Sn38,

Sk38) which are the most important features in determining the PWL slab ADF. Consistent with our and Mayer et al. (2024)'s findings, Katsuyama et al. (2023) in their +4 °C climate change experiments also found a reduction in the number of persistent weak layers in northern Japan which they interpreted as resulting from the snowpack stabilising quicker due to the higher temperatures (see also Lehning et al., 2002b).

Our results regarding the general decline of avalanche danger but also the shift of the AP away from the PWL slab and wind

slab towards the wet problem appear to be consistent with the earlier work covering Troms by Kuya et al. (2024) and Dyrrdal et al. (2020). The latter analysis of snow-avalanche related climate indices implied a higher frequency of wet-snow avalanches and slushflows in the future due to increasing snow melt and winter rain. However, the Troms climate fact sheet (Hisdal et al., 2017, later updated by Hisdal et al., 2021) pointed out that in the long run the frequency even of wet avalanches and slushflows will decline as less snow is available. Our results concur with this but indicate that while wet-snow avalanche danger will likely

indeed decline at low elevation, at higher elevation it remains considerable even at the end of the century. A similar conclusion emerged from Kuya et al. (2024)'s analysis of snow, sleet, and rain projections in Norway, indicating that in Troms at low elevation the amounts of snow will decline, while little to no decline was found at high elevation. However, they noted that despite the small changes of snow amount, the water content of the snow will increase, supporting our results of increased wet avalanche danger at higher elevation.

An elevation dependence of future avalanche development was also found in studies in other regions. Castebrunet et al. (2014) in their investigation of projections of future avalanche activity in the French Alps reported that in the northern French Alps avalanche activity declines during the 21st century, while in the southern Alps it increases, especially during winter (15 December – 15 March). They explained the discrepancy with the difference in altitude, the northern French Alps being generally lower than the southern French Alps, consistent with our findings. Notably, they also reported a 20–30 % decrease of



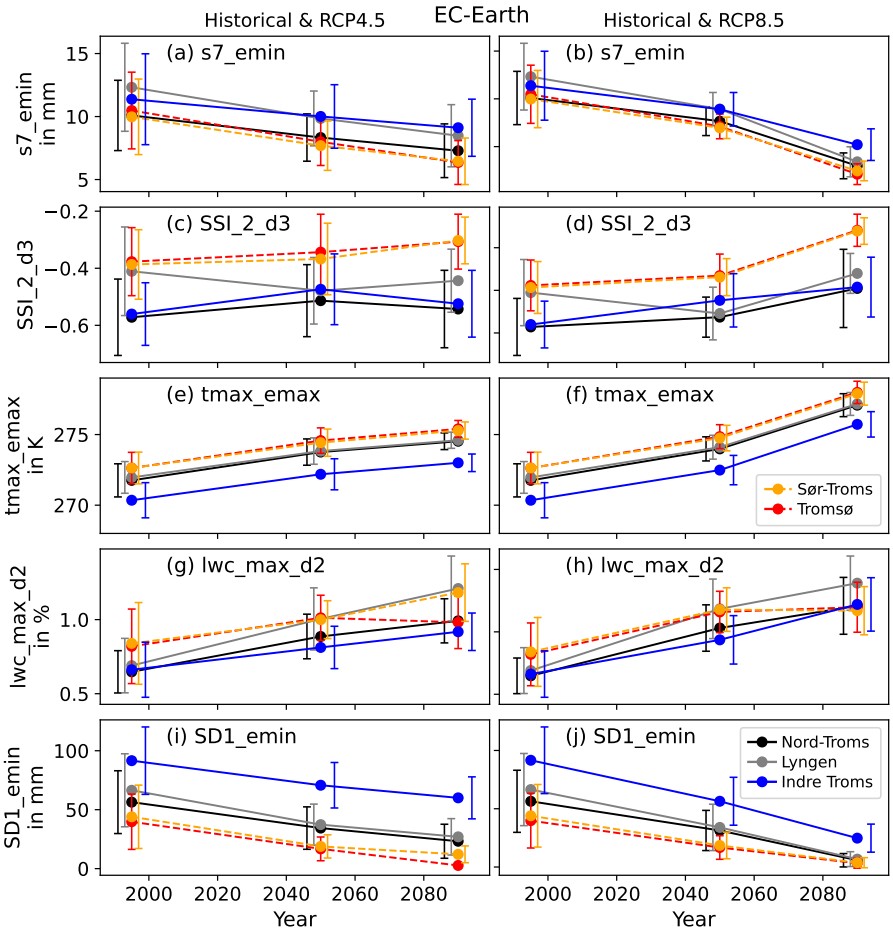

**Figure 11.** NorCP- and SNOWPACK-derived predictive features for the EC-Earth RCP4.5 (left panels) and RCP8.5 (right panels) simulations. The errorbars indicate the 1-$\sigma$ standard deviation. The errorbars are slightly horizontally shifted for better readability. For the feature definitions see section 2.5 and Tables B1 and B2 in Appendix B.

avalanche activity in general across the French Alps, qualitatively in agreement with the results presented here. Another similar finding is that there is relatively little difference between the climate scenarios by mid-century, while differences become more marked by the late century. As observed by Castebrunet et al. (2014), this implies that climate policies may have an impact on late-century avalanche danger (also compare Lazar and Williams, 2008). However, it appears that for the mid-century the impact would be marginal. The shift of the dominant AP from dry to wet was also found recently for the Swiss Alps by Mayer 555 et al. (2024). Similar to the present study and Castebrunet et al. (2014), they reported a general decline of December through May avalanche activity (10–60%), resulting mainly from less frequent dry-snow avalanches which are only partly compensated by more frequent wet-snow avalanches. An elevation dependence is also clear from Mayer et al. (2024), as stations below 2300 m a.s.l. exhibited decreasing wet-snow avalanche activity while above 2300 m wet-snow avalanches became more frequent



during the 21st century. We note that with a maximum elevation of about 1800 m Troms is generally much lower than in the
French (Castebrunet et al., 2014, separate into low, medium, and height at 1800, 2400, and 3000 m, respectively) and the Swiss
Alps (Mayer et al., 2024, divide high and low at 2300 m). This likely explains the similarities between the projected future
changes in avalanche activity in Troms and the Alps, despite the large latitudinal difference (Troms being at about 70 °N versus
the Alps at about 45 °N).

## 7 Summary and conclusions

This study investigates past and future changes of avalanche danger in northern Norway based on a machine-learning method-
ology. The danger levels, published in the daily Norwegian avalanche bulletin, for the general avalanche danger as well as three
avalanche problems (wind slab, PWL slab, wet) are aggregated from a 5-level scale to a binary scale of avalanche days (AvDs)
and non-avalanche days (non-AvDs), and the avalanche-day frequency (ADF) is interpreted as a metric to gauge avalanche
activity. Random forest (RF) models are optimised and trained to predict the AvDs based on predictive features derived from
downscaled reanalysis data (NORA3) as well as downscaled future climate projection data (NorCP). In addition, the SNOW-
PACK model is utilised to simulate the snow cover. For the historical period (1970–2024) only marginal linear trends are found,
concurring with earlier work covering northern Norway (Eiselt and Graversen, 2025). However, this finding is uncertain, in
line with previous findings stating that historical trends in avalanche activity remain unclear (Eckert et al., 2024). Considerable
decadal variability of avalanche danger is encountered, and following the earlier work by Eiselt and Graversen (2025), a link-
age is established between the Arctic Oscillation (AO) and the avalanche danger in northern Norway. However, this linkage
depends on the avalanche problem, and while the general avalanche danger as well as the wind slab and wet snow avalanches
appear partly linked with the AO, this is not the case for the PWL slab. The connection of the AO with the ADF is traced to the
most important predictive features used in the RF model. Many of the meteorological parameters that determine the general
avalanche danger, the wind slab, and the wet snow avalanche problem (AP) are significantly correlated with the AO index.
In contrast, the SNOWPACK-derived stability indices are less well correlated, hereby explaining the observed weaker linkage
of the AO and the PWL slab. For the 21st century, based on the future climate projections, the RF machine-learning model
predicts significant changes in avalanche danger, which are, however, dependent on the emission scenario, the elevation, and
the avalanche problem. For the general avalanche danger, the PWL slab, and the wind slab AP, the ADF is found to decline
throughout the 21st century. This is the case for both studied emission scenarios, but the changes are more severe in RCP8.5
than in RCP4.5. The wet snow ADF exhibits a more nuanced development which is more dependent on elevation than the
other APs. The wet snow ADF in fact increases until mid-century but this increase continues towards the late century only in
the regions at higher elevation, while in the regions at lower elevation, it declines. Hence, our study appears to concur with the
developing consensus expressed in Eckert et al. (2024) that in lower regions, avalanche frequency will likely generally decline
but in higher regions it will increase due to wet avalanche danger.

Several issues with our study should be pointed out. A concern with the Norwegian avalanche bulletin is the large extent
of the warning regions (Eiselt and Graversen, 2025). This requires a severe spatial aggregation of the meteorological and



snow data, likely obscuring information that would be important for a more spatially refined avalanche-danger prediction. The input data for the SNOWPACK model are also strongly spatially aggregated. The result that the SNOWPACK-derived parameters are found among the most important features, especially for the PWL slab, is encouraging in this regard, but

the fact that the RF model performance does not increase compared to Eiselt and Graversen (2025) raises doubts about the current level of information gain from SNOWPACK. Another issue is that SNOWPACK has not been validated for the Arctic conditions prevailing in northern Norway (see the exploratory study by van Herwijnen et al., 2024). However, work is currently ongoing for the operational implementation of the SNOWPACK model in Norway (Herla et al., 2024). A further issue is that compared to some other studies focusing on future climate projections, based on several ensemble runs of the future climate

simulations (Katsuyama et al., 2023; Mayer et al., 2024), we are restricted to the two single-ensemble runs available via NorCP. Thus, our study likely does not comprise a representative sample of possible future conditions and development of snow avalanche hazard in northern Norway. We note that more high-resolution (1 km) climate data are available for Norway from the Norwegian Meteorological Institute (Norwegian Meteorological Institute, n.d.a; Wong et al., 2016) in the form of statistical downscalings of the EURO-CORDEX dynamical downscalings for Europe (Jacob et al., 2014). These data are

unsuitable for our work because they lack wind information and their resolution is daily. Nonetheless, we have attempted to train RF models to generate ADF projections based on these data. The results (Figs. S13 to S15 in the Supplement) generally support our conclusions based on the NorCP data (for a brief discussion see Appendix A). A new version of the statistical downscaling data based on the EURO-CORDEX simulations including wind data is currently in development and will be released in October 2025 (Anita Verpe Dyrrdal, personal communication, 2025; see also Norwegian Meteorological Institute,

n.d.c; Dyrrdal et al., 2025). Employing temporal downscaling techniques, as done by, e.g. Mayer et al. (2024), may enable the utilisation of SNOWPACK also for these data and make available a large and more robust set of future climate projections for the purposes of avalanche danger prediction in Norway. As a final, more general concern we point to the notorious uncertainty of regional future climate projections. In northern Europe, the Atlantic Meridional Overturning Circulation (AMOC) is an important phenomenon influencing the regional climate (e.g. Jackson et al., 2015; van Westen et al., 2024). The AMOC is

projected to decline in most future GCM simulations conducted for CMIP, although the magnitude of the decline varies widely between GCMs (Weijer et al., 2020; Eiselt and Graversen, 2023; Madan et al., 2024). However, few future scenarios take a full AMOC collapse into account, while recent work increasingly points to this possibility (Liu et al., 2017; Ditlevsen and Ditlevsen, 2023; Dijkstra and van Westen, 2025). A collapse of the AMOC would likely have severe consequences for northern European climatic conditions, such as a strong regional cooling (e.g. Jackson et al., 2015; van Westen et al., 2024), implying

considerable impacts on the snow avalanche hazard. However, different conclusions specifically for the northern Norwegian context may be drawn from recent research by Årthun et al. (2023) who found that the Nordic Seas overturning circulation in fact strengthens in future climate simulations, thus diverging from the development at lower latitudes and acting as a stabilising factor for the AMOC. Hence, they caution that the changes in North Atlantic overturning "should not be extrapolated to the Nordic Seas and Arctic Ocean."

Reliable climate change information for natural hazards is important for planning and decision making, especially in Arctic Norway with its low population density and communities depending on easily cut off supply lines (e.g. roads without alternative



routes; see Jacobsen et al., 2016; Hovelsrud et al., 2018). The present study builds on Eiselt and Graversen (2025) to provide climate change information for the snow avalanche hazard in northern Norway, similar to previous work for other regions (e.g. Castebrunet et al., 2014; Katsuyama et al., 2023; Mayer et al., 2024). The finding that avalanche danger is generally projected

to decline is encouraging, but it is important to take note of the projected change of avalanche characteristics – the shift from dry to wet avalanches. Thus, as suggested by Mayer et al. (2024), mitigation and hazard mapping procedures should not assume stationary conditions and be revised more frequently (e.g. every 10 years). More generally, to accommodate the uncertain and complex nature of socio-environmental challenges in a changing climate with changing natural hazards, dynamic planning approaches should be developed to ensure resilience. Natural hazard projections as conducted in this study are an important

part of such approaches and work is currently under way applying the so-called "adaptation pathways" methodology (Haasnoot et al., 2013; Buurman and Babovic, 2016; Werners et al., 2021) in northern Norway with a focus on avalanche hazard to provide an adaptation strategy adequate for the changing Arctic climate (Eiselt et al., in preparation).





## Appendix A:  Future projections from EURO-CORDEX

In addition to the NorCP data, high-resolution (1 km) future projection climate data are available for Norway from the Norwe-
gian Meteorological Institute (Norwegian Meteorological Institute, n.d.a; Wong et al., 2016). These are statistical downscalings
of the EURO-CORDEX dynamical downscalings for Europe (Jacob et al., 2014). However, no wind information is provided
and the temporal resolution is daily, preventing the utilisation of these data as input for SNOWPACK. Moreover, a robust parti-
tion of the precipitation into snow and rain also appears infeasible. Despite these shortcomings we have trained RF models using
only the predictive features available from the EURO-CORDEX-based data (i.e. only temperature- and precipitation-related
features; note that the partition into snow and rain is thus necessarily done internally by the RF model), in fact obtaining a
similar predictive performance to the RF models presented in section 4 (not shown). We note that this further questions the
usefulness of the SNOWPACK-derived predictive features, at least as they are included here. The future projections of the
ADF based on the data derived from EURO-CORDEX for three models are shown in Figs. S13 to S15 in the Supplement.
They concur insofar with the NorCP-derived results (Figs. 9 and S6) as all avalanche problems except the wet snow exhibit a
decline in ADF which is stronger in RCP4.5 than in RCP8.5. However, when it comes to the wet snow problem, the EURO-
CORDEX-derived results show a strong monotonic increase throughout the 21st century. As noted in section 6.3, this illustrates
the importance of including predictive features related to the snow depth in the ML model. The wet snow problem based on
the EURO-CORDEX-derived data is mostly predicted by temperature variables and since snow-depth-related variables are
missing this implies a constant rise in the ADF concurrent with the projected temperature rise. This presents an argument for
the usefulness of the SNOWPACK simulation data as they ensure that the ML prediction results are physically reasonable,
even if they do not appear to improve the ML model performance as gauged by standard metrics for the historical period.

## Appendix B:  Predictive feature tables

Tables B1 and B2 list and give descriptions of all the predictive features considered in this study.

## Appendix C:  Model evaluation metrics

Here we briefly present the metrics that are used to gauge model performance. Table C1 shows hits (a), false alarms (b), misses
(c), and correct non-events (d), which are used to calculate the following performance metrics:

$$PC = \frac{a+d}{a+b+c+d}, \text{ the accuracy or percentage of correctly classified samples,} \tag{C1}$$

$$PR = \frac{a}{a+b}, \text{ the precison score,} \tag{C2}$$

representing the fraction of hits among the positive forecasts (i.e., hits and false alarms),

$$RC = \frac{a}{a+c}, \text{ the recall score,} \tag{C3}$$



**Table B1.** Potential predictors constructed from NORA3 meteorological data. The "nowcast day" refers to the day of publication of the avalanche danger nowcast (see section 2.1 for details). See section 2.5 for more details on the parameter definition.

| Feature name | Description |
|---|---|
| Ptot | Daily total accumulated new precipitation (mm) |
| r1, r3, r7 | Daily, 3 d, and 7 d accumulated new liquid precipitation (mm) |
| s1, s3, s7 | Daily, 3 d, and 7 d accumulated new solid precipitation (mm) |
| rh, rh3, rh7 | Daily, 3 d, and 7 d mean of relative humidity |
| t1, t3, t7 | Daily, 3 d, and 7 d mean temperature (K) |
| tmin | Daily minimum temperature (K) |
| tmax1, tmax3, tmax7 | Daily to 7 d maximum temperature (K) |
| dtr | Daily temperature range (K) |
| dtr1, dtr2, dtr3 | Diurnal cycle one to three days before nowcast day (K) |
| dtrd1, dtrd2, dtrd3 | Thermal amplitude between one to three days before and nowcast day (K) |
| ftc | Daily freeze-thaw cycle (ftc = 1) |
| pdd | Positive-degree days (7 d sum of t1 for days with t1 > 0 °C) |
| w1, w3, w7 | Daily, 3 d ,and 7 d mean wind speed (ms$^{-1}$) |
| wmax1, wmax3, wmax7 | Daily, 3 d, and 7 d maximum wind speed (ms$^{-1}$) |
| dws | Daily wind speed range (ms$^{-1}$) |
| dws1, 2, 3 | Diurnal wind-speed cycle one to three days before nowcast day (ms$^{-1}$) |
| dwsd1, 2, 3 | Wind-speed difference between one to three days before and nowcast day (ms$^{-1}$) |
| w_dir | Daily wind direction |
| dw_dir1, 2, 3 | Daily, 2, and 3 d wind direction change |
| wdrift | Drift index (w1 × s1) (ms$^{-1}$ × mm) |
| wdrift3 | Cubed drift index (w1$^3$ × s1) (m$^{-3}$s$^{-3}$ × mm) |
| wdrift_2, 3 | As wdrift but mean wind and precipitation sum over two and three days |
| wdrift3_2, 3 | As wdrift3 but mean wind and precipitation sum over two and three days |
| nsw, nsw3, nsw7 | Daily, 3 d, and 7 d mean of net short-wave radiation at surface (Wm$^{-2}$) |
| nlw, nlw3, nlw7 | Daily, 3 d, and 7 d mean of net long-wave radiation at surface (Wm$^{-2}$) |
| _emin, _emax | (as suffix) min/max across the elevation bands |

representing the fraction of hits among the positive observations (i.e., hits and misses), as well as

$$F1 = 2\frac{PR \times RC}{PR + RC}, \text{ the F1 score,} \tag{C4}$$





**Table B2.** As Table B1, but for the SNOWPACK-derived features. See section 2.4 for more details on SNOWPACK.

| Feature name | Description |
| --- | --- |
| SD1, _3, _7 | Daily, 3 d, and 7 d average of the snow depth in (cm) |
| lwc_sum, _max | Sum (_sum) and maximum (_max) over all layers of liquid water content by volume (%) |
| lwc_s_top | Sum over the top 15 cm of liquid water content by volume (%) |
| lwc_i | $= \frac{1}{0.03} \frac{\sum_{\text{layer}} h(\text{layer}) \text{LWC}(\text{layer})}{\sum_{\text{layer}} h(\text{layer})}$; liquid water content (LWC) index; $h$ is the layer height (Mitterer and Schweizer, 2013) |
| t_top | Temperature in the top snow layer (°C) |
| SSI | Structural stability index |
| Sk38 | Skier stability index |
| Sn38 | Natural stability index |
| _100 | (as suffix) indicates index is taken from layer within the first 100 cm |
| _2 | (as suffix) indicates index is taken from layer below 100 cm |
| _d1, _d2, _d3 | (as suffix) 1 to 3 d variation of the respective index |
| _emin, _emax | (as suffix) min/max across the elevation bands |

**Table C1.** Structure of the binary confusion matrix, with $a + b + c + d = N$ being the number of all events (see e.g., Sokolova and Lapalme, 2009; Wilks, 2011; Mitterer and Schweizer, 2013). AvD stands for avalanche day and non-AvD for non-avalanche day.

| | Observation positive | Observation negative |
| --- | --- | --- |
| Forecast positive | a: correct AvD | b: false alarm |
| Forecast negative | c: missed AvD | d: correct non-AvD |

being the harmonic mean of precision and recall. A macro score represents the unweighted mean of the score over all classes,
thus treating all classes equally (e.g., Sokolova and Lapalme, 2009). Further scores employed are

$$\text{FAR} = \frac{b}{a+b}, \text{ the false-alarm rate,} \tag{C5}$$

and, finally,

$$\text{TSS} = \frac{a}{c+a} - \frac{b}{d+b}, \text{ the true skill score.} \tag{C6}$$

## Appendix D: Random forest hyperparameter set

Table D1 lists the hyperparameters found to optimise the performance of the random forest models during the grid-search
procedures for the individual avalanche problems. Additionally, it contains the number of included predictive features.





**Table D1.** The sets of hyperparameters used in the random forest models. The row "Maximum number of features" refers to the number of features considered at each split in the decision trees. "log2" indicates the binary logarithm of the number of all included predictive features. The last row indicates the number of included predictive features per avalanche problem. See section 3.3 for the optimisation procedure employed to determine the hyperparemter set and the number of features.

| Hyperparameter | Wind slab | PWL slab | Wet | General |
|---|---|---|---|---|
| Number of trees | 500 | 500 | 500 | 500 |
| Maximum depth of the tree | 50 | 50 | 50 | 50 |
| Maximum number of features | log2 | log2 | log2 | log2 |
| Minimum number of samples at leaf node | 15 | 30 | 30 | 10 |
| Minimum number of samples for each split | 15 | 30 | 30 | 10 |
| Number of included predictive features | 20 | 50 | 50 | 15 |



*Code and data availability.* The programming language Python was used to perform the data analysis and generate the figures. The random forest model was generated using the Python library scikit-learn (Pedregosa et al., 2011). The maps were produced with the library Cartopy (Met Office, 2010 - 2024). The code for downloading and preprocessing the data, as well as generating the machine-learning models and the
figures is published on Zenodo (Eiselt, 2025a), as are the optimised and trained machine-learning models and the predictive features (Eiselt, 2025b).

*Author contributions.* KUE and RGG conceived and designed the study. Material preparation, data collection and analysis were performed by KUE. The first draft of the manuscript was written by KUE with comments and revisions from RGG.

*Competing interests.* The authors have no relevant financial or non-financial interests to disclose.

*Acknowledgements.* The research is conducted under the IMPETUS project (www.climate-impetus.eu, last access 20 August 2025). This project was funded by the European Union's Horizon 2020 research and innovation programme under grant agreement No. 101037084. This work uses data from the NorCP project, which is a Nordic collaboration involving climate modeling groups from the Danish Meteorological Institute (DMI), Finnish Meteorological Institute (FMI), Norwegian Meteorological Institute (MET Norway) and the Swedish Meteorological and Hydrological Institute (SMHI) (Wang et al., 2024). We are grateful to Oskar Landgren for his help in acquiring the NorCP data. The
authors thank Debmita Bandyopadhyay, Konstantinos Christakos, and Christopher D'Amboise for valuable suggestions. Finally, we thank Stefania Munaretto for bringing the relevance of the AMOC for this topic to our attention.



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
