# Peer review of "Past and future changes in avalanche problems in northern Norway estimated with machine-learning models"

_EGUsphere, 2025_

## Author Comment (AC1)

Review of Past and future changes in avalanche problems in northern Norway estimated with machine-learning models

By Kai-Uwe Eiselt and Rune Grand Graversen

**Summary**

This paper uses a model chain to predict the past and future avalanche hazard in northern Norway. This work builds on previous work by the same authors, who developed Random Forest models to predict avalanche danger. The model chain they developed primarily consists of a dynamic downscaling of climate models in Norway for the past and future, which serves as input to the snow cover model SNOWPACK. Then, they build Random Forest models to predict avalanche days for several avalanche problems using meteorological variables (from the downscaled climate models) and snow instability variables from SNOWPACK. They show different historical trends in the frequency of avalanche days for different avalanche problems (e.g., wet, storm, wind, or persistent), as well as correlations with the Arctic Oscillation (AO). They conclude with projections of avalanche problems using climate projections (RCP4.5–8.5) for Norway, demonstrating similar results to those found in the Alps (Switzerland and France).

The paper is generally well written, well thought out, and is worthy of publication in The Cryosphere. The only major concerns I have regarding the methodology relate to the spatial aggregation of the downscaled climate simulations. In addition, more details should be provided concerning the SNOWPACK modeling for reproducibility purposes. It may also be beneficial to add a dedicated section in the discussion about the limitations and biases of their study, and how these affect their results (small one in the conclusion). There are a few punctuation issues across the text, and addressing them would enhance the flow of the manuscript.

We thank the reviewer for considering our manuscript and for the constructive review. We are grateful for the comments and provide a point-for-point response below.

As a general response regarding tenses, we prefer a style where our current work leading to the results presented in the article is referred to in present tense, while earlier work (by us or other authors) is referred to in past tense. We are of the opinion that this increases clarity and hope that it is acceptable to maintain this in our manuscript.

**Major Comments:**

1. Climate simulations tend to "smooth" extreme events due to their coarse resolution. In addition, the spatial aggregation of the climate simulations further enhances this smoothing effect, which is critical for avalanche problem types such as storm and wind slab. I believe this important bias needs to be addressed in the discussion, as it could affect the interpretation of the projected trends for these two avalanche problems. While the projected climate captures "thermal" avalanche problems such as persistent weak layers (PWL) and wet snow reasonably well, the projections for storm and wind slabs should be interpreted with caution. I also think that more information on the spatial aggregation would help the reader better understand this effect.

We agree with the reviewer that the spatial aggregation of the data is a major issue with our study. However, note that we differentiate only between the binary level non-AvD (danger level 1&2) and AvD (danger level 3&4&5), likely in itself representing a sort of "smoothing" of extreme-event-levels 4 & 5 by their combination with the more moderate level 3. Moreover, the avalanche warning regions in Norway are large (our five regions are 3000-9000 km$^2$), presenting a challenge to spatial aggregation and again likely representing in itself a "smoothing." We think this is a more general problem impacting machine-learning model accuracy, and we discuss this in the manuscript (lines 590-593). Following the suggestion in the reviewer's general comment, we will add a new section to the discussion ("6.4 Limitations") where we will include the points noted here as well as additional concerns (see the responses to the comments to line 257 and 597-598). We will also move our existing discussion of the limitations and concerns from the conclusions section (lines 590-624 in the manuscript) to this new section.

2. More details are needed concerning important parameters, parameterizations, and the simulation setup of the SNOWPACK model, in order to improve the reproducibility of this study.

We will expand the information given about our SNOWPACK setup and change the first paragraph in section 2.4 to:

"To obtain more detailed information on the snow cover, we run version 3.6.0 of the physics-based, multi-layer model SNOWPACK (Bartelt and Lehning, 2002; Lehning et al., 2002a, b). The model solves the governing conservation equations (for mass, energy, and momentum) within the snowpack and simulates the snow cover one-dimensionally. We run SNOWPACK with a 15 min computation time step, Neumann boundary conditions at the snow–atmosphere boundary, and the bucket scheme approach (Bartelt and Lehning, 2002) to simulate liquid water flow through the snow and soil. For atmospheric stability, the simplified Richardson-number correction is applied.

For soil evaporation the standard resistance approach is employed and fitted values are used for soil thermal conductivity."

However, regarding reproducibility, we note that the full SNOWPACK setup scripts are published on Zenodo (https://doi.org/10.5281/zenodo.17277192; see also the Code Availability section).

3. The figure sizes should be adjusted, as they are currently too small, and the font style does not match that of the manuscript.

We will adjust the figure sizes to increase readability. However, there is no requirement in *The Cryosphere* style guide (https://www.the-cryosphere.net/submission.html) for the font of the figures to match the font of the text (the requirement is only that the font of the figures be consistent). We are not familiar with such a requirement from other journals either and our previous article in *The Cryosphere* (Eiselt and Graversen, 2025, https://doi.org/10.5194/tc-19-1849-2025) uses the same font styles as here. Thus, we would like to maintain the font as is.

4. Several punctuation marks are missing throughout the text, which limits the flow and the comprehension of some sentences. I've highlighted a few examples below, but please check this consistently throughout the manuscript.

We are grateful to the reviewer for pointing out the punctuation issues and we will fix them in the next version of the manuscript.

**Specific comments (line number)**

**Section 1 - Introduction**

15: Already have impact the occurrence in the arctic, especially in mass movements. they are several references in the literature.

We will reformulate this to:

"Changes in climatic conditions, as observed for the 20th and as projected for the 21st century, impact the occurrence and character of natural hazards (Hock et al., 2019)."

Hock et al. (2019) is the chapter "High Mountain Areas" in the "IPCC Special Report on the Ocean and Cryosphere in a Changing Climate" which (among other things) gives a general assessment of the impact of past and future climate change on natural hazards in mountainous areas.

**Section 2 - Data**

115: Change apply to past tense "applied"

Our standard is to present our current work in present tense and earlier work (by us or other authors) in past tense. Since this "apply" here refers to our work for the current article, we would prefer to leave it as is.

158: What is slab snow??

This refers back to "new": "*new* loose and slab snow," both of which, as can be seen in, e.g., Fig. 3, are part of the avalanche problems. We will adapt this to "new loose and new slab snow," to increase clarity.

160-161 : I think a ref to Figure 3 would be great here, as I struggle the get what the number means unless I look at Figure 3.

The reference to Figure 3 appears in line 157 and the text following this reference is meant as a discussion of this figure. However, we agree that a further reference to Fig. 3 is helpful here and will add it: "Figure 3 further shows that the general ADL..."

Figure 3: is ADL on the x axis the general? Please define.

This is correct. We are grateful to the reviewer for pointing out the oversight and will add this information in the figure caption. In addition, we will further improve the consistency by colouring the left y axis red and the right y axis black as in Figs. 1 and 2.

183: punctuation is needed to enhance the flow between danger and we.

Changed.

201: punctuation is needed to enhance the flow between conditions and Lind.

Changed.

205: too-strong is a bit vague for an amount of precipitation, or maybe it is about precipitation rate? Please clarify.

We will change this to "too-high amount of winter precipitation"

205-209: Not sure the relevance of these information to describe the dataset, it feels more like an introduction, or maybe as a part of the discussion to compare with the results.

We thank the review for bringing this to our attention and agree that this should not be part of the description of the data. It will be removed in the next version of the manuscript.

213: punctuation is needed to enhance the flow between cover and we.

Changed.

216: not sure if this is the right reference for key summary of SNOWPACK. This paper is an update status on snow cover modeling in avalanche forecasting including CROCUS and SNOWPACK.

We thank the reviewer for pointing this out. Indeed, Morin et al. (2020) gives a good summary of the key points of the *operational* setup of SNOWPACK, which does not necessarily correspond to our setup. Thus, we will remove this sentence. Please see our response to Major Comment 2 above for the new first paragraph of section 2.4.

219: punctuation is needed to enhance the flow between temperature and we.

Changed.

220-221: punctuation is needed to enhance the flow between (TSS) and we.

Changed.

226: Do you end up with 4 SNOWPACK simulations per warning region? Each simulations have the average grid cell for 4 elevation band? Is 20 the total number per warning region or the entire study area? A sentence that summarizes how many simulations per warning region is needed.

In line 227 we will reformulate and add.:

"…, assuming flat terrain, leading to four SNOWPACK simulations per warning region. This means that for our whole study area of Troms county 20 SNOWPACK simulations are performed…"

230-235: Maybe reduce these lines to one or two sentences, as it limits the comprehension of your methods. We assumed that it is included and it complicates for nothing this section.

We will shorten these lines somewhat. Unfortunately, we are unsure what the reviewer means with "We assumed that it is included,…" Please also refer to our response to the reviewer's next comment below.

257: why explain this? Either remove it or put it into the result.

We are somewhat confused by this comment. One of the major concerns the reviewer states in the general comment is the spatial aggregation of the data. In the mentioned lines we report that we have performed a sensitivity analysis specifically regarding the spatial aggregation of the data. That is, we have tested different ways of spatial aggregation, and this had little to no impact on our results. However, to be clearer, we will make this into its

own paragraph, partly reformulating it, and add another statement about a further sensitivity test we have performed in the meantime:

"To investigate the impact of the strong spatial aggregation on the prediction accuracy we have tested several different ways of spatial aggregation or selection of grid cells. We have generated the predictors for ten specifically wind- and snow-exposed grid cells per avalanche region (SNOWPACK was run for these grid cells specifically as well, see section 2.4), but this did not improve the performance of the machine-learning models. Moreover, we have tested taking the maxima/minima of the features for each individual elevation band (resulting in a much larger number of features), but the impact on model performance was again minimal."

Note that this paragraph will be moved to the new section on the limitations of the study (section 6.4).

258: based, use past tense .

As described above, we prefer a style where we use present tense for the work we do for the present article while earlier work is referred to in past tense. Also note, that in, e.g., line 256 we use past tense for something we have done ("We have tested...") but here we refer to something that did not become part of the paper (here this refers to the analysis based on a different kind of spatial aggregation), which is why we find past tense appropriate here.

**Section 3 - Methods**

264: you need to state at least the main analysis and parameter we should not need to read another paper.

We are unsure what the reviewer means with "main analysis" here. We recognise that we should have referred to Table E1 here, which lists the precise hyperparameter settings used for the random forest model. We will adjust section 3.1 in the following way:

"To establish the statistical linkage between meteorological data and avalanche danger we employ the widely used random forest (RF) model (Breiman, 2001), which 'grows' a number of decision trees (DTs, Breiman et al., 1984) that 'vote' on the final prediction result. Like Eiselt and Graversen (2025) we use the RF implementation from the Python library scikit-learn version 1.3.0 (https://scikit-learn.org/, last access 23 September 2025). One RF model is trained for each avalanche problem, resulting in four different RF models. The hyperparameter setups for the individual RF models are presented in Table E1 in Appendix E. The data were split into a test (winter 2020/21 and 2022/23) and training (remaining winters) data set."

We hope that from this the reader gets all necessary information from the paper itself and does not need to read our previous work.

265: do you have values or maybe a figure to show the imbalance and the effect of the algorithm.

We will add a figure about the class imbalance in the Supplement. We reproduce the figure here as Fig. 1.

[Figure]

*Figure 1: Class imbalance of the training and test data. The number of avalanche days (AvDs) and non-avalanche days (non-AvDs) is shown for the original imbalanced data in black and for the balanced data in red. The balancing was performed with the SMOTE algorithm (see section 3.2).*

283: should the F1 score gives that?

We are not fully sure what is meant here, but the F1 score is just one way to aggregate precision and recall score. We wanted to consider several metrics that give more information about the prediction results of our model.

Figure 4: Please adjust the font to match the manuscript, and define what is general? Maybe remove true danger, as danger bring confusion between danger level and avalanche problem.

Regarding the font style please refer to our response to Major Comment 3 above.

We will remove "danger" from the axis labels since we agree with the reviewer that this is confusing. We thank the reviewer for pointing this out. Also, "General" refers to the "General ADL". We will improve the consistency of this throughout the article.

Section 4 – Model Performance and features importances.

This section also results like section 5.

While it is true that this section also effectively contains results, it is more of an evaluation of the model and we wanted to frame our results mainly in terms of the title of the article, i.e., as the past and future avalanche problems in Troms.

310: the false alarm is also very high.

This is true, but since 100 % minus RC essentially is the false alarm, it would be redundant to state this.

313: please stick to one definition either problem or danger level.

We thank the reviewer for pointing this out and we will increase the consistency on this by using "problem" instead of "danger". We will change this throughout the manuscript (see the tracked changes version).

**Section 5 - Results**

Section 5.1.1 : please use past tense.

As mentioned earlier, we prefer a style where our current results are presented in present tense while earlier results are presented in past tense.

339 - 340 : please rephrase this sentence.

Without knowing what the reviewer's issue with this sentence is, it is difficult to accommodate this comment. Nevertheless, we will rephrase the sentence to the following:

"We expand the correlation analysis of Eiselt and Graversen (2025) between the general ADF and the AO index by considering the individual avalanche problems."

343 : maybe refer to the figure 8.

Added.

344 : be consistent with fig. Or figure.

We are unsure what the reviewer means here. If this refers to the apparent inconsistency between using "Figure 8" in line 340 and "Fig. 8" in line 343, we want to point out that the style guide to *The Cryosphere* (https://www.the-cryosphere.net/submission.html) says:

'The abbreviation "Fig." should be used when it appears in running text and should be followed by a number unless it comes at the beginning of a sentence, e.g.: "The results are depicted in Fig. 5. Figure 9 reveals that…".'

361 : was this define in the method section.

We are unsure what the reviewer is referring to here. We assume this comment to be a question similar to the following:

"Was this [the Monte-Carlo simulation method] defined in the methods section?"

If this is indeed the question, then the answer is no, this was not defined in the methods section, since we thought that a Monte-Carlo simulation to test the significance of the difference between two samples is a well-known method, that does not require a dedicated explanation. However, we will add a more detailed description in the appendix as follows:

"Appendix B: Monte-Carlo significance test

The significance of the difference between the avalanche-day frequency (ADF) predicted for the different periods within a given future climate scenario (Fig. 9 and Figs. S9–13 in the Supplement) is tested with a Monte-Carlo simulation which is described in the following: Let A and B be two periods (all periods comprise 20 avalanche seasons, i.e., 20 ADF values). The 'observed' statistic is calculated as the mean ADF of climate period A minus the mean ADF of period B. Then period A and B are combined and their ADF values are randomly permuted. The permuted values are subsequently divided again into two periods ($A_{test}$ and $B_{test}$) and the difference between their mean values is calculated. If this mean difference between $A_{test}$ and $B_{test}$ is larger than the mean difference between A and B a counter is incremented. This procedure is repeated for a given number of permutations (here 100,000). Finally, the p value is calculated by dividing the counter by the number of permutations, giving the fraction of instances in which a random shuffling of the data produced a larger difference than the real difference of scenarios A and B, i.e., the observed statistic."

For transparency we note here that we noticed an error in another instance where we applied a Monte-Carlo simulation, which means that we will remove the sentence (lines 379-380): "However, the difference is statistically significant (p < 0.05) only in Indre Troms (based on a Monte-Carlo simulation)."

Section 5.2: there is way more reference to supplemental figures than figure 9, please put these into the text. Figure S9 has more references than figure 9. Or maybe the appendix, which is more accessible.

The frequent references to the supplementary figures only occur to support our statements regarding the statistical significance of the changes. It appears to us that the best way to accommodate the reviewer's comment is to simply remove most of the individual references, since we already state at the beginning of section 5.2 that the significance of the differences is shown in the supplementary figures.

**Section 6 - Discussion**

Section 6.1: how the precision of the model affects your results especially the PWL.

Partly in response to this comment as well as to reviewer #2, we have performed a further sensitivity analysis, where we trained additional random forest models by excluding different years as test data. We briefly describe this in the new section 6.4 on the limitations of the study and provide a more extensive analysis in a new text and new figures in the Supplement. We find that while the predicted absolute values of the ADF are different between the different random forest models, the changes over time are consistent across models. Thus, we are confident in our conclusions.

419 - 428: I think it might be worth it to discuss these factors between the development and the trigger of the PWL.

When it comes to the meteorological factors, we generally only consider short-term (up to 7 days) changes or maxima/means/minima, and thus we do not discuss the longer-term evolution between development and trigger of the PWL here. We will try to make this clearer in the text. We believe the longer-term evolution to be represented by the SNOWPACK-derived parameters and stability indices.

491: would it be better yrs instead of y.

Changed to "yr".

497: it might also be warmer and thaw events stabilizing the snowpack.

We thank the reviewer for pointing this out. We will add it as a point in this line. The new paragraph will be:

"Three points may be noted regarding the potential linkage of the AO with the PWL slab AP: (1) The positive correlation of the AO index with temperature may imply thaw events that stabilise the snowpack, leading to fewer PWLs. (2) The generally negative correlation of the PWL slab ADF with the AO index (Fig. 10) may result from the concurrent higher wind slab and wet ADF, reflecting the fact that fewer weak layers persist for a long time as avalanches readily release due to frequent new snow and wind-drift loading. (3) The performance of the RF model in terms of predicting PLW slab ADF is low (Fig. 4), calling into question the robustness of the results regarding this AP (see also section 6.4)."

**Section 7 – Summary and conclusions**

593: why not write meteorological input as both are spatially aggregated for input to the rf's model.

This sentence will be moved the new section 6.4 on the limitations of the study and thus be slightly reformulated anyway.

597-598: I think this is rather concerning. it was also point out that SNOWPACK struggle to model artic snowpack, because of the high thermal gradient (Domine et al., 2019).

We are grateful for the suggestion of this article, and we will include it as a reference in our manuscript. However, we want to point out that Domine et al. (2019) test the SNOWPACK model for a location (Bylot Island) with a climate rather different from the climate in the county of Troms, due to the generally milder climate in northern Europe compared with North America. For example, according to Domine et al. (2019), the average temperature on Bylot Island is -14.5 °C while in Tromsø it is 2.6 °C, according to Hisdal et al., (2021). Other towns in Troms county are, e.g., Finnsnes with 3.4 °C, Harstad with 4.0 °C and Bardufoss with 1.0 °C. This indicates that the issues found by Domine et al. (2019) may not be as severe in Troms County as they are on Bylot Island. Similarly, the study by van Herwijnen et al. (2024) that we cite in the manuscript shows that the difference between the snowpack in Troms and in the Alps is smaller than that between Alaska and the Alps.

We note that the Norwegian Water and Energy Directorate (NVE) has recently published a new version of SNOWPACK specifically for the Norwegian conditions, but this was too late for our article.

We will include these points in our new section on the limitations of our study (section 6.4).

References

Domine, F., Picard, G., Morin, S., Barrere, M., Madore, J. B., & Langlois, A. (2019). Major issues in simulating some Arctic snowpack properties using current detailed snow physics models: Consequences for the thermal regime and water budget of permafrost. *Journal of Advances in Modeling Earth Systems*, *11*(1), 34-44.

---

## Author Comment (AC2)

General comments:

This paper addressed past and future avalanche frequencies in northern Norway using the SNOWPACK model and the random forest (RF) model. The target avalanches were not only general problems but also wind slab, persistent weak layer slab, and wet snow avalanche problems. The past avalanches were investigated mainly with consideration of their linkage to the Arctic Oscillation (AO) index, and the avalanche frequencies were well correlated with the AO index. The future dry-snow avalanches would be estimated to decrease, while the wet-snow avalanches would increase until mid-century. The topic and results are valuable for the scientific community. The introduction provided a nice review of the global warming impact on avalanches.

However, I have a concern about the originality of this study. I agree with the authors that this work presents an original case to show future avalanche problems in Norway; however, the other aspects of originality seem limited. The random forest model used had mainly been developed in the authors' previous work. The linkage between avalanches and the AO index had also been found in the authors' previous work. The future estimations, including their procedure, are similar to those in previous works, such as Mayer et al. (2024). I feel that the originality of this work would be insignificant for "The Cryosphere", even though the differences in locations themselves are valuable to the scientific community.

We thank the reviewer for considering our manuscript and for the detailed points. We provide a point-for-point response to the comments below.

We appreciate the reviewer's concerns regarding the novelty of our work, as we feel it is important to address that clearly in the manuscript. The random forest model in our previous work was only developed to be applied to the general avalanche danger and here we develop new models for the individual avalanche problems. In fact, since the danger levels for the individual avalanche problems were not directly available, it was necessary to convert the raw data (size, sensitivity, distribution) from Varsom into danger levels for each avalanche problem and to analyse the problems to find out which of them to focus on (this analysis is presented briefly in section 2.1; see also Fig. 3). Also, we changed our procedure for the preparation of the features to attempt to include more of the underlying variability potentially influencing avalanche danger (as described in section 2.5). Moreover, we utilised a new and different model optimisation procedure (section 3.3). We note that in the manuscript we clearly indicated our departure and further development relative to earlier work. However, we will attempt to make this even clearer, e.g. already in the abstract. Further, while the linkage between avalanche danger in northern Norway and the AO was established in our previous work, again, this was only done for the general avalanche danger, while here we investigate the connection with individual avalanche problems.

Regarding the future projections and the similarities with the work of Mayer et al. (2024), we recognise that our work is conceptually similar, although our application is for a different region having fundamentally different climate and meteorology than that of Mayer et al. (2024). However, as we point out in the article, the data situation in Norway is substantially different from Switzerland and it required considerable additional developments (as detailed in the article) to arrive at our results. Also, we think that it is interesting and encouraging in itself that our results agree with Mayer et al. (2024) and this should not be used as an argument that our work lacks novelty.

The utilization of the RF model also seems problematic. From my understanding, the authors estimated the avalanche-day frequency (ADF) by cumulating the daily 1/0 output from the RF model. However, this procedure might lead to a biased ADF because the RF model was not optimized by minimizing the error of the ADF. Actually, the sum of predicted AvD for wind slab avalanches is 440, while that of true AvD is 245 (Fig. 4), indicating a mean bias towards overestimation in the ADF. I suppose the RF model should be a regression type, rather than a binary type. I recommend confirming the RF model's reproducibility regarding ADF by comparing it to the observation.

We thank the reviewer for pointing out the bias in the model predictions. We will include a new figure in the Supplement that shows the comparison of the distribution of true and predicted AvDs and non-AvDs to make this even clearer. We appreciate the suggestion of using a regression-type RF model, but this would involve a whole new feature aggregation and selection as well as model optimisation procedure and a fully new analysis since the ADF is a seasonally aggregated metric, while our metric (AvD) is a daily metric. Hence, this is outside the scope of this study and is left for future work. However, we have conducted a new sensitivity analysis to account for the bias described by the reviewer. Since this sensitivity analysis also accommodates the following comment, we describe it further below.

This may be related to the above problem, but I am also concerned that the authors did not consider uncertainties arising from the RF model. Seeing Fig. 4, the RF model may produce a very large uncertainty in its projection. For example, the RF model incorrectly predicts general avalanches with probabilities of 36% in AvD predictions and 17% for non-AvD predictions (Fig. 4). I am not certain, but the uncertainty range is comparable to or more than that of climate models. Furthermore, the authors converted AvD/non-AvD from avalanche danger level simply by a threshold (Section 2.1), which also causes uncertainty. However, the authors show no data to discuss this kind of uncertainty arising from the conversion. These problems would change the results of statistical tests for linear trends in past and future avalanche frequencies (Figs. 6, 7, 8 ,9), and if so, the authors' conclusion

may be changed. Authors should quantitatively demonstrate the uncertainties associated with past and future projections arising from the RF model, and these uncertainties should be considered in the statistical analysis. This point is crucial for ensuring the reliability of the RF models' estimation.

We appreciate the concerns of the reviewer about uncertainty, and, as mentioned above, we have conducted a new sensitivity analysis to account for this uncertainty. However, please first note that our purpose here is to show the *tendency* of the development of avalanche danger and we are interested in the *change* of the ADF and not its absolute values, and this is especially true when it comes to the linear trends, the AO-ADF linkage, and the future development. As a general note, this is also what is typically done in climate model studies of the future (i.e., the focus on the *anomaly* instead of on the absolute value) since climate models struggle with representing the real-world climate state. This problem is hence not specific to our study and constitutes a general issue with model-based research on climate change. Thus, we believe many of the concerns of the reviewer, while definitely not irrelevant, are at least somewhat overstated. Accordingly, our new sensitivity analysis also focuses on the *tendencies* and not the absolute values. We utilise the fact that when training our random forest model we excluded two years (winters ending in 2021/23) as test data and train three additional models excluding different years (winters ending in 2018/22, 2019/24, 2020/22). We perform the historical and future-projection ADF predictions with the new random forest models and compare the results (including those from the random forest model analysed in detail in the manuscript) to investigate the robustness of our conclusions. While, as expected, the absolute ADF values vary between the models, the tendencies (linear trends, AO-ADF correlation, future development) are similar across models, increasing our confidence in our conclusions. We will include the description of this new sensitivity analysis in our new section 6.4 on the limitations of our study and add a more detailed analysis as well as several new figures to the Supplement to show the comparison between the different random forest models.

Finally, we are unsure how we should "show data" to discuss the uncertainty arising from our *convention* of using the AvD for danger level 3 and larger. Again, this is just a *convention* to facilitate our study on the *tendency* or *change* of the avalanche danger. Please note that (as also noted by reviewer #3) the danger levels do not necessarily correspond to a probability of avalanche occurrence across time and space and, hence, a quantitative uncertainty analysis appears inappropriate here. For convenience, we reproduce our justification for our AvD/non-AvD convention from our earlier work (Eiselt and Graversen, 2025) here:

"Furthermore, the ADF appears related to avalanche activity, since Pérez-Guillén et al. (2024a) in a case study in the Swiss Alps using an automated seismic avalanche detection system found that

on days with no avalanche, the mean ADL was 1.9±0.8, while on days with at least one avalanche, it was 3.2±0.5, hence providing a clearly binary appearance. Similarly, in an investigation of Swiss backcountry GPX tracks as a proxy for non-avalanche events, Techel et al. (2024) found that for non-events the median probability of ADL ≥ 3 was only 0.14, while for events it was 0.58. Hence, on a day with ADL 3 or 4, avalanche events are likely, while they are unlikely on days with ADL 1 or 2, justifying our definition of AvD and non-AvD."

Specific comments:

L41: "RPCs" seems to be a typo instead of "RCPs".

We thank the reviewer for noting this oversight.

L46: You need to define the abbreviation NorCP here.

Again, thank you for spotting this oversight.

L55: From my understanding, Lazar and Williams (2008) assessed a potential avalanche period very simply based on air temperature exceeding 0 °C or not. Although I do not want to treat authors' opinions carelessly, I disagree with this.

We thank the reviewer for this assessment, and we agree with it. We will add a caution about this simplified approach by Lazar and Williams (2008) in our text, changing the beginning of the paragraph to:

"To the authors' knowledge, Martin et al. (2001) was the first study investigating the change of avalanche activity under changing climatic conditions based on a statistical linkage between meteorological parameters and avalanches. By implementing constant positive perturbations of temperature and precipitation in their statistical model, they found for a study area in the French Alps that while new-snow avalanches declined, wet-snow avalanches increased in frequency (at least relatively). Lazar and Williams (2008) were likely the first to employ future emission scenarios to investigate future development of avalanche activity or danger, although they simply defined their avalanche periods based on a temperature threshold."

L105–121: These contents are better moved to Section 2.

We thank the reviewer for this suggestion. However, we view this as a motivation and a very short introduction to our study and thus would like to keep it in the introductory section of our article.

L133: A dual abbreviation definition of ADL.

We will remove the definition here.

L136: What are the active avalanche problems?

We will indicate that these problems are the ones relevant for that specific day. We will change "active" to "relevant".

L140: What are distribution and sensitivity?

We will extend this part of the sentence to:

"…, the latter being derived from the spatial distribution of hazardous sites and the sensitivity to triggers (Müller et al., 2016a, b; Statham et al., 2018; Müller et al., 2023)."

We hope this makes clear what is meant by distribution and sensitivity.

Figure 3: Is the left axis showing the number of avalanche days? What is the avalanche problem frequency?

We will change the caption of this figure. Please note that further changes in the caption were necessary to accommodate a comment by reviewer #1:

"Average danger level per avalanche problem (AP; red) and the number of days on which the AP was identified by the forecasters (black) in northern Norway. The average danger level was calculated only for the days the specific AP was identified. The ADL (avalanche danger level) on the x axis refers to the general avalanche problem. The data cover the period from winter 2016/17 to 2024/25 for the general avalanche problem and 2017/18 to 2024/25 for the other APs."

We will also change the figure slightly: The red colour will be moved to the left to be consistent with the other figures and individual avalanche problems on the x axis will be reordered according to the frequency of occurrence.

Section 2.4: Please describe the model settings for soil.

We will add the settings we have used although we did not change the default.

Section 2.4: How did you calculate liquid water content (LWC)? LWC is very important for wet avalanches (Fig. 5). Furthermore, local LWC exceeding 5% is very important for wet-avalanche predictions (Wever et al. 2016). This point should be taken into account.

We are unsure what the reviewer exactly means here. We use the standard LWC output from SNOWPACK. The parametrsiation used to simulate liquid water flow through the snow is the bucket scheme. We will add this information to section 2.4

Section 2.4: Please describe how you obtain daily snowpack variables. The original output of the SNOWPACK model is generally hourly data, but you use daily avalanche data.

Please note that this is precisely described in section 2.5, the whole purpose of which is the description of the procedure we utilise to obtain the features that represent the input for the random forest model. More information is also given in Table B2 in the appendix which lists the features derived from the SNOWPACK output and gives a brief description of each of them.

L218: How do you prepare long-wave radiation data?

We do not use any long-wave radiation data as this is not required by the model if one gives the surface temperature. We provide the link to the SNOWPACK documentation where this is stated: https://snowpack.slf.ch/doc-release/html/requirements.html

L218: You used the net short-wave radiation. So, you mean that the albedo depends on a land surface model implemented in a meteorological model? If so, does this affect the SNOWPACK simulation? The snowpack calculation is very sensitive to the short-wave radiation.

Yes, this is a weakness of our data input as we do not have the incoming and outgoing components of the net radiation available. However, given the large-scale spatial aggregation of our data, we caution against overinterpreting the influence of this on the SNOWPACK calculation.

L220: The linear model should be described in the Appendix or Supplement.

We will add a short description of the model as well as the model equation and a new plot showing the relationship of predicted and true TSS to the Supplement.

L226: How did you calculate precipitation, wind, and relative humidity? A simple arithmetic mean is generally inappropriate for these variables.

The data were spatially averaged by simple arithmetic mean (see section 2.5). While this may be "inappropriate" as the reviewer states, we also performed several sensitivity tests, trying different aggregations as described in the lines below. Since this did not have any noticeable impact on our results, we decided to stay with the arithmetic mean approach.

L228–235: These lines should be described in the Appendix or Supplement.

In response to reviewer #1 as well as the last comment below we will add a dedicated section on the limitations of our study (section 6.4). We will move these lines there. Please note that the point of including these lines was to anticipate a comment such as the previous one and to defend our choice of the averaging methodology we use.

Section 2.5: This content is too hard for readers without a background in the RF model. Can you merge this content into Section 3?

Maybe there is some kind of misunderstanding here and we are unsure what the reviewer means. The content of section 2.5 in itself has nothing to do with the random forest model and is simply a description of how we aggregate the raw input data from NORA3, NorCP, and SNOWPACK to obtain daily values.

L273: What are min_samples_leaf, min_samples_split, max_depth, n_estimators, and max_features?

We thank the reviewer for pointing out our oversight of not explaining the hyperparameters. We will change this sentence to:

"This variation mostly derives from the hyperparmeters min_samples_leaf (MSL) and min_samples_split (MSS), representing the number of samples that remain at a leaf and after a split, respectively. The other hyperparameters (see Table E1 in Appendix E) appear to have a much smaller influence (not shown). We note that MSL and MSS have a similar effect, both determining the number of samples at the leaves of the DTs in the RF. In fact, MSL is the "finer" tuning parameter and MSS has no impact if MSS ≤ MSL. Given these hyperparamerter dependencies we deviate from the model optimisation and feature selection procedure conducted in earlier work (e.g. Pérez-Guillén et al., 2022, Hendrick et al., 2023, Eiselt and Graversen, 2025); that is, we do not perform a randomised grid search over all the different hyperparameters, but instead only test different values of MSL, while holding the other hyperparameters constant (Table E1 in Appendix E)."

Note that, as described in this new formulation, we have made the decision to change the procedure slightly and now only vary MSL since MSS has no effect if it is smaller than MSL. This required an update of almost all figures, since we re-optimised and re-trained the random forest model. However, this resulted only in small changes in the details of the analysis, which may be seen in the tracked changes document.

We will also change our description of the random forest in section 3.1:

"To establish the statistical linkage between meteorological data and avalanche danger we employ the widely used random forest (RF) model (Breiman, 2001), which 'grows' a number of decision trees (DTs; Breiman et al., 1984) that 'vote' on the final prediction result."

L285: You mean leave-one-out cross-validation? However, your procedure is not the leave-one-out cross-validation, but the k-fold cross-validation, actually. Leave-one-out cross-validation is a method in which a single independent data point is excluded from the training data. In this study, a single independent data is a 1/0 in a day, not a year.

We thank the reviewer for pointing this out and we will change it in the text, clarifying that one "fold" corresponds to one winter. We were rather referring to the fact that we always leave one winter out, i.e., it is a leave-one-winter-out validation.

L286: I do not understand why five years of training data are available even though you have Norwegian avalanche bulletin's data from 2017/18 to 2024/25.

We apologise for the lack of clarity here: As is typical for machine-learning training, we spilt our data in a training and test data set. We exclude two seasons (winter 2020/21, and 2022/23) as test data, leaving five years of data for training. We will add this information in section 3.1.

L509–512: This is also problematic from the viewpoint of the applicability of RF models to future climate. Does the RF model linearly increase the wet-snow ADF by increasing air temperature (or liquid water content) if only there were enough snowpack? However, one of the necessary conditions for wet avalanches is a high liquid water content, locally exceeding 5% (Wever et al., 2016). Satisfying this condition, wetting of an initially below-freezing snowpack is important (Mitterer et al. 2011). Capillary barriers or melt–freeze crusts are also key phenomena. Therefore, the authors need to confirm whether the models' behavior in linearly increasing wet-snow ADF by increasing air temperature is really appropriate in Norway.

We thank the reviewer for these suggestions and for pointing out the important references. However, since the random forest model methodology is rather a black box and (as can be seen in Table D1 in Appendix D) grows 500 decision trees of depth 50 which is impossible for the human mind to comprehend, it is infeasible to perform the kind of investigation suggested here. Please note that there may be a misunderstanding here: The random forest model does not predict the ADF itself. It rather predicts if a certain day is an avalanche day or if it is not and it uses thresholds to do this. Thus, a random forest model is certainly not linear. The only thing we can say is that the temperature is an important feature that has some explanatory power for the wet avalanche problem. In preliminary work after the manuscript was submitted, we attempted to use the Shapley value methodology (as was done by Pérez-Guillén et al., 2025, for Switzerland) to get more information on how the individual features affect the random forest model prediction, but this only tells us that generally higher values of the feature t_max (i.e., daily maximum temperature) make it more likely that a wet avalanche day is predicted, while lower values make it less likely. Thus, such a detailed analysis may be possible with a much simpler decision tree model, maybe of depth 3 or 4, meaning most of the features we use would need to be excluded. Something like this was done by, e.g., Mitterer and Schweizer (2013). But given the severe spatial aggregation of our data and the large areas of our warning regions we are unsure of the utility of such exact thresholds. However, it may be an interesting avenue for future research.

References:

Mitterer, C. and Schweizer, J.: Analysis of snow-atmosphere energy balance during wet-snow instabilities and implications for avalanche prediction, The Cryosphere, pp. 205–216, https://doi.org/10.5194/tc-7-205-2013, 2013.

Pérez-Guillén, C., Techel, F., Volpi, M., and van Herwijnen, A.: Assessing the performance and explainability of an avalanche danger forecast model, Nat. Hazards Earth Syst. Sci., 25, 1331–1351, https://doi.org/10.5194/nhess-25-1331-2025, 2025.

L590–637: These lines should be described in Section 6.

We thank the reviewer for the suggestion. The reviewer #1 had a similar comment, and we will move these lines to our new section "6.4 Limitations".

References:

Mayer, S., Hendrick, M., Michel, A., Richter, B., Schweizer, J., Wernli, H., and van Herwijnen, A.: Impact of climate change on snow avalanche activity in the Swiss Alps, The Cryosphere, 18, 5495–5517, https://doi.org/10.5194/tc-18-5495-2024, 2024.

Wever, N., C. Vera Valero, and C. Fierz (2016), Assessing wet snow avalanche activity using detailed physics based snowpack simulations, Geophys. Res. Lett., 43, 5732–5740, doi:10.1002/2016GL068428.

Mitterer C, Hirashima H, Schweizer J. Wet-snow instabilities: comparison of measured and modelled liquid water content and snow stratigraphy. Annals of Glaciology. 2011;52(58):201-208. doi:10.3189/172756411797252077